# MOZ/ENL complex is a recruiting factor of leukemic AF10 fusion proteins

Yosuke Komata[1,5], Akinori Kanai [2,3,5], Takahiro Maeda [4], Toshiya Inaba[3] & Akihiko Yokoyama [1] ✉

Changes in the transcriptional machinery cause aberrant self-renewal of non-stem hematopoietic progenitors. AF10 fusions, such as CALM-AF10, are generated via chromosomal translocations, causing malignant leukemia. In this study, we demonstrate that AF10 fusion proteins cause aberrant self-renewal via ENL, which binds to MOZ/MORF lysine acetyltransferases (KATs). The interaction of ENL with MOZ, via its YEATS domain, is critical for CALM-AF10-mediated leukemic transformation. The MOZ/ENL complex recruits DOT1L/AF10 fusion complexes and maintains their chromatin retention via KAT activity. Therefore, inhibitors of MOZ/MORF KATs directly suppress the functions of AF10 fusion proteins, thereby exhibiting strong antitumor effects on *AF10* translocation-induced leukemia. Combinatorial inhibition of MOZ/MORF and DOT1L cooperatively induces differentiation of CALM-AF10-leukemia cells. These results reveal roles for the MOZ/ENL complex as an essential recruiting factor of the AF10 fusion/DOT1L complex, providing a rationale for using MOZ/MORF KAT inhibitors in *AF10* translocation-induced leukemia.

Acute leukemia is a major cause of death in children with cancers. Our understanding of the underlying molecular mechanisms of leukemia and the availability of effective treatment methods have not met the current medical needs. Alterations in the genes encoding epigenetic regulators, such as lysine methyltransferases (KMTs), including MLL (also known as KMT2A), and lysine acetyltransferases (KATs), including MOZ (also known as KAT6A and MYST3) and MORF (also known as KAT6B, MYST4, and QKF), cause leukemia by conferring stem cell-like unlimited self-renewal properties to non-stem hematopoietic progenitors[1,2].

AF10 (also known as MLLT10) and ENL family proteins, including ENL (also known as MLLT1) and AF9 (also known as MLLT3), are major fusion partners of MLL[3]. AF10 and ENL family proteins form a complex with DOT1L KMT, which methylates histone H3 lysine 79[4,5]. The DOT1L complex co-localizes with the MLL KMT complex and other ENL-containing transcriptional regulators, including the AF4 family/ENL family/P-TEFb (AEP) and MOZ/MORF KAT complexes, at the promoter-proximal transcribed regions of transcriptionally active genes[6–8]. ENL was not found as a complex component of MOZ/MORF in the early biochemical purifications of stably assembled complexes[9], suggesting that ENL is a transient interactor that binds to MOZ/MORF in a context-dependent manner[7]. MOZ/MORF and HBO1 (also known as KAT7 and MYST2) constitute a subgroup of functionally similar KATs termed the MYST family. HBO1 physically associates with the MLL complex and co-localizes with those ENL-containing protein complexes[10,11]. These epigenetic and transcriptional regulators comprise a transcriptional activation system (hereafter referred to as the MLL/MOZ/AEP-mediated transcriptional activation system), which activates a range of CpG-rich promotors through multiple cell divisions[12]. During embryonic development, CpG-rich promoters are

[1]Tsuruoka Metabolomics Laboratory, National Cancer Center, Tsuruoka, Yamagata 997-0052, Japan. [2]Laboratory of Systems Genomics, Department of Computational Biology and Medical Sciences, Graduate School of Frontier Sciences, the University of Tokyo, Kashiwa, Chiba 277-0882, Japan. [3]Department of Molecular Oncology and Leukemia Program Project, Research Institute for Radiation Biology and Medicine, Hiroshima University, Hiroshima, Hiroshima 734-8553, Japan. [4]Division of Precision Medicine, Kyushu University Graduate School of Medical Sciences, Fukuoka, Fukuoka 812-8582, Japan. [5]These authors contributed equally: Yosuke Komata, Akinori Kanai. ✉e-mail: ayokoyam@ncc-tmc.jp

regulated by polycomb group (PcG) and trithorax group (TrxG) complexes[13,14]. This MLL/MOZ/AEP-mediated transcriptional activation system plays a major role in the maintenance of gene expression during embryonic development and in the self-renewal of hematopoietic stem cells during normal hematopoietic development[15–18].

Gene rearrangements of *AF10* generate diverse AF10 fusion proteins with many partners besides MLL, including CALM (also known as PICALM), DDX3X, NUP98, and XPO1 (also known as CRM1), causing multiple lineages of malignant leukemia, including acute myeloid (AML), B-cell acute lymphoblastic (B-ALL), and T-cell acute lymphoblastic leukemia (T-ALL)[19–23]. Because the AF10/DOT1L complex co-localizes with MLL and MOZ at CpG-rich promoters and shares ENL as a component with MOZ and AEP complexes[7], we hypothesized that AF10 fusions might promote their oncogenic activity by overactivating the MLL/MOZ/AEP-mediated transcriptional activation system, similar to the functions of MLL and MOZ fusions.

In this work, we investigated the structural and functional requirements of AF10 fusions for oncogenic transformation in murine disease models. This is the first study to reveal the ENL/MOZ complex as an essential co-factor for AF10 fusion protein complex recruitment to target promoters. We suggest that the ENL/MOZ complex can serve as a potential therapeutic target in *AF10* translocation-induced leukemia.

## Results

### AF10 fusions transform hematopoietic progenitors via ENL

Aberrant self-renewal induced by overactivation of the MLL/MOZ/AEP-mediated transcriptional activation system can be evaluated by an ex vivo myeloid progenitor transformation assay[7]. In this assay, oncogene-transduced c-Kit-positive hematopoietic progenitor cells (HPCs) that are harvested from the bone marrow (BM) of healthy mice are cultured in semi-solid media in the presence of myeloid cytokines[24]. Cells transduced with a functional oncogene typically exhibit upregulated *Hoxa9* expression in the first-round colonies as well as vigorous colony-forming activity in the third and fourth rounds of replating (Fig. 1a and Supplementary Fig. 1). The structural requirements of MLL-AF10 were the minimum targeting module (MTM: PWWP domain of LEDGF and CXXC domain of MLL) that binds to the target chromatin[25], THD2 domain of MLL that recruits an HBO1 complex[10], and OM-LZ domain of AF10 that binds to DOT1L HMT[5,8]. An artificial fusion of MTM and the THD2 domain (hereafter referred to as MTMT) tethered to the full-length DOT1L, instead of the OM-LZ domain, transformed HPCs (see MTMT-DOT1L in Fig. 1a). The MTMT-DOT1L constructs with mutations deficient in KMT activity (i.e., MTMT-DOT1L dKMT) or the ability to bind ENL (i.e., MTMT-DOT1L dMISD) failed to transform, thereby indicating that both KMT activity and ENL association are necessary for MLL-AF10-mediated leukemic transformation. Replacement of the OM-LZ domain by the full-length ENL also caused a full transformation, highlighting the importance of ENL association (see MTMT-ENL in Fig. 1a). A minimum fusion protein (i.e., MTM-AHD) comprising MTM and the ANC1 homology domain (AHD) of ENL, which binds to the AEP components, transformed HPCs[25], indicating that the tethering of AEP to the MLL target chromatin is the central mechanism of overactivation of the MLL/MOZ/AEP-mediated transcriptional activation system[12]. These results suggested that MLL-AF10 provides ENL to the MLL target chromatin to recruit AEP[8,10].

The CALM-AF10 fusion transforms immature HPCs in mice[26] and causes T-ALL, B-ALL, and AML in humans[27]. The minimum structural requirements of CALM-AF10 are the nuclear export signal (NES) of CALM and the OM-LZ domain of AF10 (see NES-AF10´ in Fig. 1a)[26,28,29]. Artificial gene constructs of ENL fused with CALM or NES (see CALM-ENL or NES-ENL in Fig. 1a) transformed HPCs (although the NES-DOT1L fusion did not), indicating that association with ENL is essential in CALM-AF10-mediated leukemic transformation. As wild-type ENL did not transform HPCs, NES appeared to confer the gain-of-function to

ENL to aberrantly activate *Hoxa9*. Immunoprecipitation (IP) followed by western blotting (WB) in DOT1L-deficient cells demonstrated that the NES-AF10´ fusion interacted with ENL in a DOT1L-dependent manner (Fig. 1b). Transplantation of NES-ENL-transduced HPCs resulted in leukemia in vivo, which was recapitulated in the secondary recipients (Fig. 1c). These results indicated that CALM-AF10 transformed HPCs via ENL.

Next, we assessed various other AF10 fusions including DDX3X-AF10 and NUP98-AF10[23,30], using the ex vivo myeloid progenitor transformation assays. Both DDX3X-AF10 and NUP98-AF10 transformed HPCs via the OM-LZ domain, which was interchangeable with ENL (Fig. 1a and Supplementary Fig. 1). RNA-sequencing (RNA-seq) analysis of three independently established immortalized cells (ICs) with each AF10 fusion type revealed that they expressed a similar gene set including *HoxA* genes, which is different from that in the originating HPCs (i.e., c-Kit+ BM; Fig. 1d and Supplementary Fig. 2a–c). Other differentially expressed genes in c-Kit + -BM and AF10 fusion-ICs are the genes implicated in the lysosome, hemopoiesis, and lipid biosynthesis (Supplementary Fig. 2d, e). CALM-AF10- and NES-ENL-ICs tended to have near-identical expression profiles, which was also similar to those of MLL-AF10-ICs, while NUP98-AF10´- and DDX3X-AF10´-ICs expressed slightly different gene sets, which included the genes implicated in MAPK cascades, suggesting that the activation status of proliferative signaling pathways may be different among the AF10 fusion subtypes (Supplementary Fig. 2c, f, g). These results suggest that AF10 fusions transform hematopoietic progenitors by activating similar gene sets including *HoxA* genes via ENL.

### MOZ recruits ENL and DOT1L to *HOXA* gene promoters

We examined the genomic localization of ENL and other functionally related components using chromatin immunoprecipitation (ChIP) coupled with deep sequencing (ChIP-seq)[7,10]. In HEK293T cells, ENL was co-localized at promoter-proximal regions with the DOT1L, MOZ, and AEP complexes (e.g., AF4) and HBO1[10] (Fig. 2a and Supplementary Fig. 3a). ENL-target promoters were enriched with RNA Polymerase II (RNAP2) with a non-phosphorylated CTD heptapeptide motif (RNAP2 non-P) bound by MOZ[7]. The ChIP signal intensity of ENL was positively correlated with that of the MOZ, DOT1L, AEP, and HBO1 complexes, sharing common target genes (Supplementary Fig. 3b, c). These results suggest that ENL functionally collaborates with MOZ, DOT1L, AEP, and HBO1 at promoter-proximal regions.

Next, we generated MOZ- and DOT1L-deficient HEK293T clones (two biological replicates each; Fig. 2b) and analyzed them using WB, reverse transcription (RT) coupled with quantitative polymerase chain reaction (qPCR), and ChIP-qPCR analyses. The MOZ-deficient clones exhibited reduced *HOXA5* and *HOXA9* expression levels (Fig. 2c). In addition, MOZ is necessary for the presence of ENL and DOT1L at the *HOXA* loci, but not at the *MYC* locus, as shown by a genome-wide analysis of the localizations of ENL and DOT1L in MOZ-deficient cells (Fig. 2d). Reduced ENL ChIP signal was uniquely seen at a subset of ENL target genes including *HOXA* genes (Fig. 2e and Supplementary Fig. 3d). Furthermore, ChIP signals of DOT1L were severely reduced at the *HOXA* loci as well by MOZ knockout (Fig. 2f and Supplementary Fig. 3d). These results suggest that *HOXA* genes, but not all ENL-target genes, are regulated dependently on MOZ. Moreover, ChIP-qPCR analysis confirmed that the chromatin occupancy of DOT1L and ENL was profoundly reduced at the *HOXA* loci in MOZ-deficient cells (Fig. 2g), indicating that MOZ is required for ENL and DOT1L recruitment at the *HOXA* gene promoters. Additionally, the DOT1L-deficient clones demonstrated reduced levels of *HOXA5* and *HOXA9* expression (Fig. 2c) and the localization of ENL at the *HOXA* loci (Fig. 2g), suggesting that the continued presence of the DOT1L/ENL complex is required for sustained *HOXA* gene expression. The chromatin occupancy of MOZ was moderately reduced (30–50%) in DOT1L-deficient cells, suggesting a positive feedback

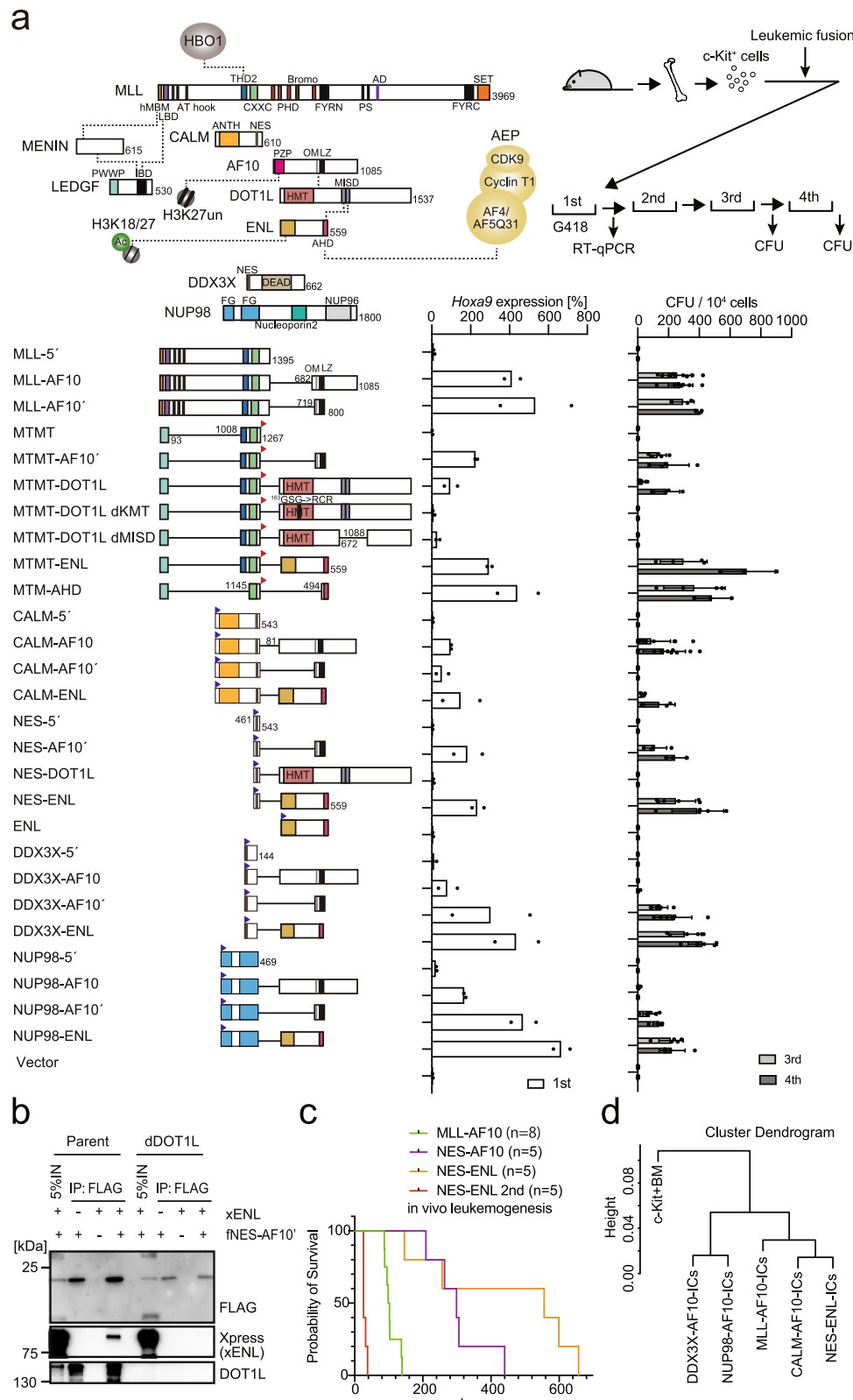

loop. These results indicated that the MOZ/ENL complex recruits DOT1L to *HOXA* gene promoters.

## NES-ENL fusion binds to the target gene promoters via the YEATS and AHD domains of ENL

We evaluated the structural requirements of NES-ENL-dependent leukemic transformation by the ex vivo myeloid progenitor transformation assay (Fig. 3a and Supplementary Fig. 4a). Deleting the AHD domain resulted in complete loss of transformation (see NES-dAHD in Fig. 3a), indicating that the interaction with AHD binders (e.g., AF4 or DOT1L) is required for transformation. Unlike the MLL-ENL fusion (see MTM-AHD in Fig. 1a), deletion of the non-AHD portion of ENL resulted in a loss of transformation (see NES-AHD in Fig. 3a). Moreover, a loss-of-function mutation of the YEATS

**Fig. 1 | AF10 fusions transform hematopoietic progenitors via ENL. a** Structures required for AF10 fusion-mediated myeloid transformation. Various AF10 fusion constructs were examined to assess their transforming ability of myeloid progenitors. HA-tag (indicated by red triangles) was fused to MTM and MTMT constructs. FLAG-tag (indicated by blue triangles) was fused to other AF10 fusion constructs. Dotted lines indicate protein-protein interaction. A schema of a myeloid progenitor transformation assay is shown at the top. *Hoxa9* expression normalized to *Gapdh* in first-round colonies (left) is shown as the relative value of CALM-AF10 (arbitrarily set at 100%) (mean of two biological replicates). Colony-forming ability at the third- and fourth-round passages (right) is shown with error bars (mean ± SD of biological replicates, $n \geq 3$). **b** Association of NES-AF10 fusion with ENL in the presence or absence of DOT1L. IP-western blotting (WB) analyses of the chromatin fraction of HEK293T cells [the parental clone or a DOT1L-knockout clone (dDOT1L)] transiently expressing the FLAG-tagged (indicated as f) NES-AF10′ fusion (fNES-AF10′) construct and Xpress-tagged (indicated as x) ENL (xENL) were performed. Co-purification of ENL was observed only in the presence of DOT1L. **c** Leukemogenesis by NES-ENL fusion in vivo. Various AF10 fusion-derivatives including NES-ENL were transduced to c-Kit-positive hematopoietic progenitors and transplanted into syngeneic mice. Primary NES-ENL leukemia cells were harvested from the bone marrow (BM) and transplanted into recipient mice. **d** Hierarchical clustering analysis of RNA-seq profiles of AF10 fusion-ICs. Normalized count data in various AF10 fusion-ICs was clustered using R ward D2 method. Source data are provided as a Source Data file.

domain[7,31,32] (i.e., Y78A) resulted in a loss of transformation, suggesting that the YEATS-mediated interaction with acetylated histones or MOZ/MORF KATs is required for CALM-AF10-mediated leukemia. An artificial construct of AF9 fused with the NES (i.e., NES-AF9) transformed HPCs. These results suggest that ENL and AF9 are commonly associated with transformation. NES of CALM has been reported to interact with XPO1, a nuclear export receptor also known as CRM1[33]. An artificial fusion of XPOI and ENL activated *Hoxa9* and immortalized HPCs, suggesting that NES-ENL activates *Hoxa9* expression via interaction with XPO1. In addition, stable expression of NES-ENL in HEK293T cells resulted in elevated expression of *HOXA5* and *HOXA9* (Fig. 3b, c). This indicates that NES-ENL is a gain-of-function mutant of ENL that can activate *HOX* genes in both HPCs and HEK293T cells via a common mechanism.

Next, we examined ENL-mediated protein interactions in the chromatin fraction using the fanChIP method, which captures protein-protein interaction in the chromatin fraction[34]. The Y78A mutation abrogated the association of NES-ENL with MOZ, whereas the AHD deletion resulted in a loss of interaction with DOT1L and Cyclin T1 (Fig. 3d). MOZ-ENL interaction was detected in DOT1L-deficient cells (Supplementary Fig. 4b), suggesting that MOZ/ENL complex formation occurs independently of DOT1L/ENL complex formation. Because the defective mutations in the YEATS and AHD domains showed no transformation activity in HPCs (Fig. 3a), both of the associations with MOZ and DOT1L seem essential for NES-ENL-mediated transformation. In addition, ChIP-qPCR analysis of HEK293T cell lines expressing various ENL constructs showed that wild-type ENL was associated with the promoter regions of *HOXA5* and *HOXA9*, similar to CALM-AF10 (Fig. 3e), indicating that ENL can target these promoters. Furthermore, NES-ENL exhibited a similar targeting ability, whereas defective mutations in the YEATS and AHD domains completely abrogated it. Thus, the stable association of ENL with target chromatin requires YEATS- and AHD-mediated interactions. Taken together, chromatin targeting by CALM-AF10 is mediated by the MOZ/ENL complex, whose complex formation depends on the YEATS domain.

## ENL, DOT1L, and MYST family KATs are required for CALM-AF10 leukemia cells

To examine whether ENL, DOT1L, and MYST family KATs are required for CALM-AF10 leukemia cell proliferation, we performed CRISPR/Cas9-mediated competition assays on murine and human leukemia cells. Like MLL-ENL-ICs, the proliferation of CALM-AF10-ICs depended on both *Enl* and *Dot1l* (Fig. 4a). *Moz* knockout quickly eliminated CALM-AF10-ICs, while it also attenuated the proliferation of MLL-ENL-ICs. *Morf* knockout did not dramatically affect the proliferation of CALM-AF10-ICs or MLL-ENL-ICs, suggesting that MOZ is the predominant ortholog in these cell types. These tendencies were consistently observed using multiple different sgRNAs (Supplementary Fig. 5). Moreover, Brpf1, a common component of MOZ/MORF complexes, was also required for both cell types. HBO1 is required for both CALM-AF10 and MLL-ENL, as it is generally required for the survival of leukemia stem cells[11,35]. Similarly, the proliferation of P31/FUJ, a human

leukemia cell line harboring CALM-AF10 fusion, depended on ENL and DOT1L (Fig. 4b). In this cell line, MORF, but not MOZ, was more critical for proliferation and appeared to be the dominant isozyme. HBO1 is essential for both P31/FUJ and HB1119, a human leukemia cell line harboring MLL-ENL. Taken together, ENL, DOT1L, and MOZ/MORF KATs are essential for CALM-AF10-mediated leukemic transformation.

## Inhibition of MOZ/MORF-mediated acetylation induces dissociation of MOZ/ENL and CALM-AF10 complexes from target chromatin

To further dissect the role of MOZ/MORF-mediated acetylation in ENL-mediated functions, we examined the effects of MOZ/MORF inhibition on CALM-AF10-bearing leukemia cells (i.e., P31/FUJ) using WM1119, a MOZ/MORF KAT inhibitor[36]. A reduction in bulk acetylation of histone H3K14/23 (H3K14/23ac) was observed in P31/FUJ cells after WM1119 treatment (Fig. 5a). WM1119 completely inhibited the proliferation of P31/FUJ cells, while it slowed down proliferation (but failed to induce complete differentiation) of *MLL* translocation-induced leukemia cell lines (i.e., HB1119, MV4-11, and MonoMac-6) and Nucleophosmin mutation (NPMc)-bearing leukemia cells (i.e., OCI-AML3) at 10 μM (Fig. 5b and Supplementary Fig. 6a). WM3835, another MYST KAT inhibitor for MOZ/MORF KATs with more profound inhibitory effects on HBO1[11], had similar anti-proliferative effects on P31/FUJ cells, but not on HB1119, MV4-11, and OCI-AML3 cells (Supplementary Fig. 6b), despite effective inhibition of histone H3K14 acetylation (Supplementary Fig. 6c). These results indicate that MLL leukemia cells are critically dependent on the presence of MOZ/MORF/HBO1 proteins but not their KAT activity. EPZ-5676, a DOT1L KMT inhibitor, had anti-proliferative effects on both P31/FUJ and *MLL* translocation-induced leukemia cell lines similar to those reported in previous studies in other disease models (Fig. 5a, b)[37,38]. Neither WM1119 nor EPZ-5676 affected the proliferation of K562 cells (Supplementary Fig. 6a). From these results, we conclude that CALM-AF10 leukemia cells are uniquely dependent on the acetylation activity of MOZ/MORF KATs.

To elucidate the mechanism of action of these inhibitors, we cultured HEK293T cells in the presence of WM1119 for 5 days, which substantially decreased bulk H3K14/23ac (Supplementary Fig. 6d). This was observed as early as 24 h post-treatment (Fig. 5c). The *HOXA5* and *HOXA9* expression levels were also reduced by WM1119 treatment (Fig. 5d). In addition, ChIP-qPCR analysis showed that the presence of ENL, MOZ, and DOT1L was reduced at the *HOXA5* and *HOXA9* loci (Fig. 5e), suggesting that MOZ/MORF-mediated acetylation is implicated in the recruitment and retention of MOZ- and DOT1L-complexes to the target chromatin. IP-WB analysis of the chromatin fraction of HEK293T cells that expressed FLAG-tagged ENL showed that WM1119 treatment reduced the amount of ENL in the chromatin fraction but did not attenuate MOZ/ENL complex formation (Fig. 5f). These results suggest that chromatin recruitment or retention of the MOZ/ENL complex is hampered by MOZ/MORF KAT inhibition. Consequently, CALM-AF10 recruitment was reduced by WM1119 treatment (Fig. 5g). Treatment with EPZ-5676 effectively reduced di-methylation of

 

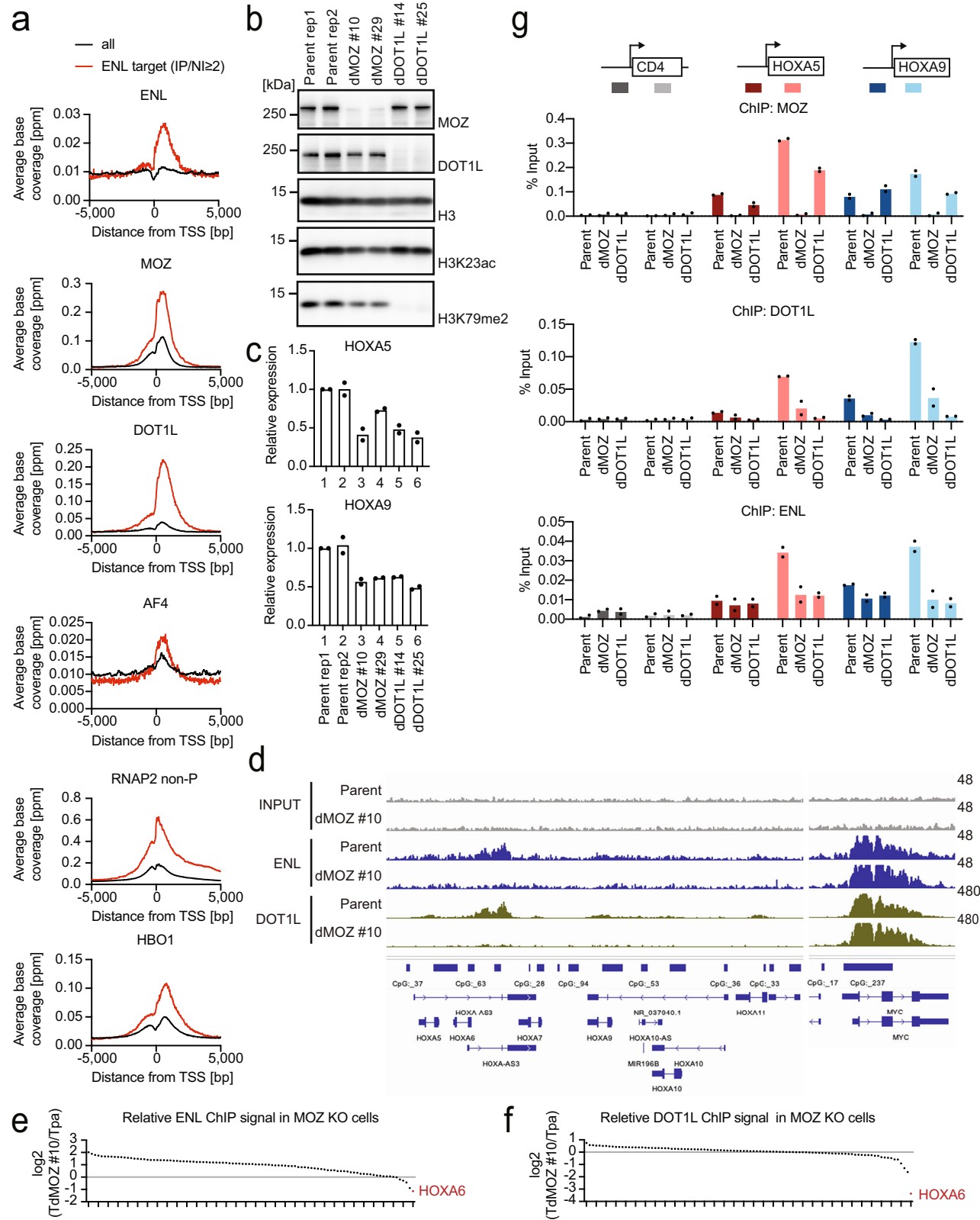

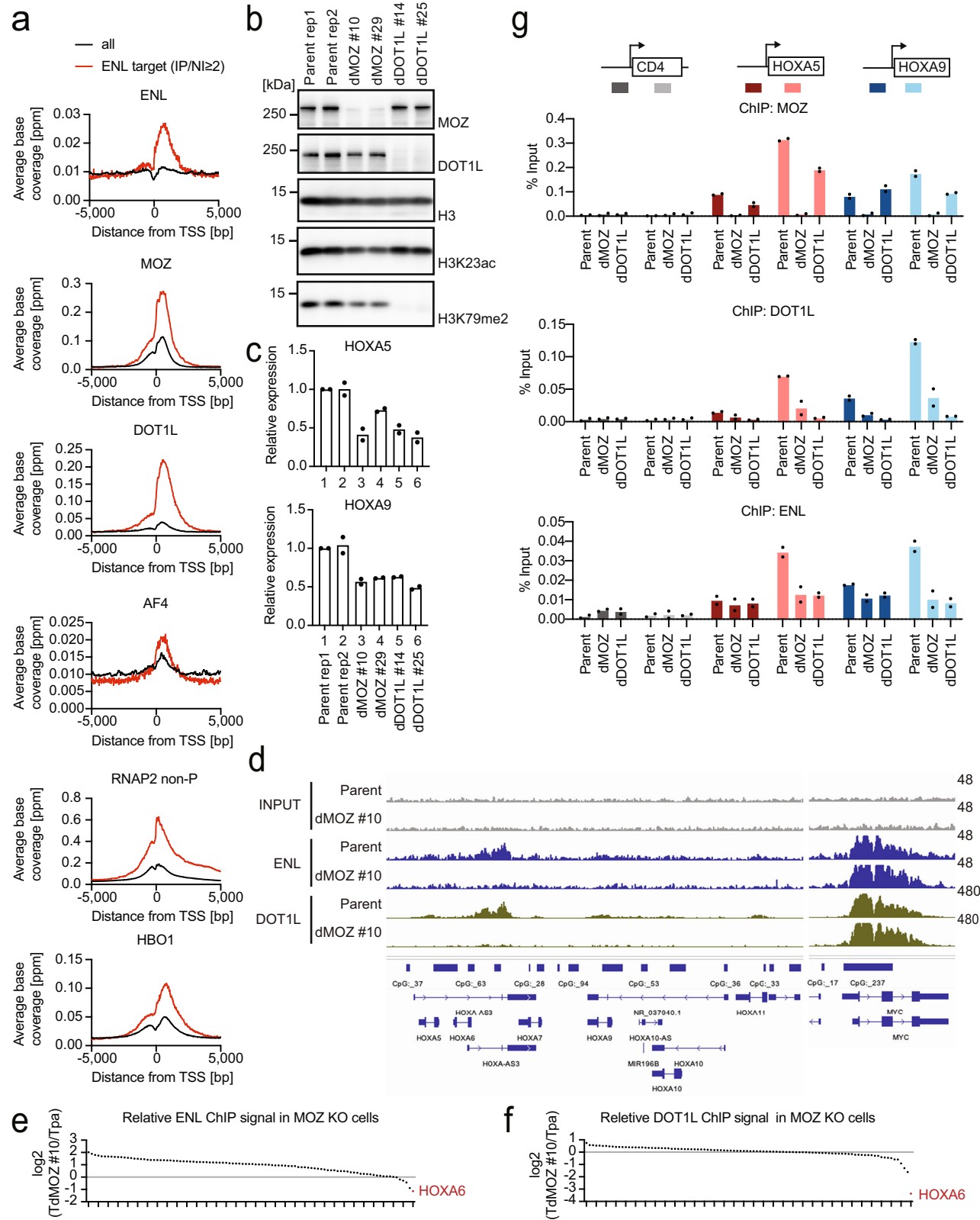

histone H3 lysine 79 (H3K79me2; Supplementary Fig. 6d), lowered *HOXA5* and *HOXA9* expression in HEK293T cells (Fig. 5d), and reduced the chromatin retention of MOZ and ENL (Fig. 5e). These results suggest that both MOZ/MORF-mediated lysine acetylation and DOT1L-mediated lysine methylation maintain the presence of MOZ- and DOT1L-complexes at the *HOXA* loci, promoting CALM-AF10-mediated leukemia.

## Pharmacological inhibition of the MOZ/MORF KATs suppresses CALM-AF10-mediated leukemia in vivo

We tested the antitumor effects of the MOZ/MORF KAT inhibitor on CALM-AF10 leukemia cells in vivo. First, P31/FUJ cells were transduced with a modified luciferase gene (LUC2Red) and transplanted into immunocompromised mice. After engraftment was confirmed by measuring luciferase-mediated bioluminescence using an in vivo

**Fig. 2 | MOZ recruits ENL and DOT1L to *HOXA* gene promoters. a** Genomic localization of various transcriptional regulators in HEK293T cells. ChIP-seq analysis of the chromatin of HEK293T cells for the indicated proteins. The average ChIP signal distribution of the indicated proteins around the transcriptional stat sites (TSSs) is shown for the ENL-target genes (red) or all genes (black). ChIP-seq tags of ENL and its input chromatin at all genes were clustered into a 2-kb bin (0 to +2 kb from the TSS) and the genes with the ENL ChIP/input ratio ≥2 were defined as ENL target genes. **b** Protein expression of MOZ and DOT1L and histone modification levels. WB of the of HEK293T cells was performed in two technical replicates of the parental clone and two biological replicates of MOZ- or DOT1L-knockout clones each. **c** *HOXA* gene expression in MOZ and DOT1L-deficient cells. RT-qPCR was performed on the HEK293T clones shown in b (mean of two technical replicates). **d** Localization of ENL and DOT1L in MOZ-deficient cells. ChIP-seq analysis was performed on MOZ knockout HEK293T cell line and its parental clone. ChIP-signals were visualized using the Integrative genome browser (IGV, Broad Institute). **e** Relative ChIP signals of ENL between a MOZ-deficient cell line and its parental clone. ENL ChIP signals of the dMOZ#10 clone in the bins of 0–1 kb of the top 100 ENL-bound genes of its parental clone were subdivided by those of the parental clone. HOXA6 is highlighted in red. **f** Relative abundance of DOT1L ChIP signals of the top 100 DOT1L-bound genes in a MOZ-deficient clone is shown as in e. **g** Localization of MOZ, DOT1L, and ENL. ChIP-qPCR was performed on two technical/biological replicates of each genotype for the indicated gene loci using qPCR probes designed for the pre-TSS (−1 to −0.5 kb of TSS) and post-TSS regions (+1 to +1.5 kb of TSS) of each gene. ChIP signals are expressed as a percentage of the input with error bars (mean of two technical/biological replicates). Source data are provided as a Source Data file.

imaging system (IVIS), WM1119 was administered at 50 mg/kg three times a day for 2 weeks (Fig. 6a). The treatment markedly suppressed leukemia cell expansion and prolonged the survival of the recipient mice.

We then examined the effects of MOZ/MORF inhibition on murine HPCs immortalized with a variety of AF10 fusions. WM1119 exhibited a substantial anti-proliferative effect on CALM-AF10-, DDX3X-AF10´-, and NUP98-AF10´-ICs ex vivo and induced complete differentiation in most cell lines (Fig. 6b, c). It also induced the differentiation of MOZ-TIF2-ICs, which is dependent on the intrinsic KAT activity of MOZ-TIF2 (Supplementary Fig. 7a)[7,39]. In contrast, the treatment only moderately slowed the proliferation of MLL-ENL-ICs (Fig. 6b). MLL-AF10-ICs exhibited a range of anti-proliferative behaviors (Supplementary Fig. 7a), suggesting that the MOZ/MORF KAT inhibitor is generally effective for *AF10* translocation-induced leukemia but less effective for *MLL* translocation-induced leukemia, in which MLL fusion proteins are recruited to the target chromatin via the MLL modules (Figs. 5b and 6b). The IC50 of CALM-AF10-ICs in terms of colony formation was 12 nM, compared to 12 µM in MLL-ENL-ICs (Supplementary Fig. 7b, c). The DOT1L inhibitor suppressed the proliferation of CALM-AF10-ICs ex vivo, with a stronger anti-proliferative effect on CALM-AF10-ICs than MLL-ENL-ICs, consistent with the relatively fast kinetics of the attenuation of proliferation induced by *Dot1l* knockout in CALM-AF10-ICs (Fig. 4a).

We further examined the antitumor effects of WM1119 on CALM-AF10-mediated leukemia in an immunocompetent context. Primary leukemia cells harvested from moribund mice were transduced with LUC2Red ex vivo and transplanted into secondary syngeneic recipients (Fig. 6d). WM1119 treatment effectively suppressed leukemogenesis in an immunocompetent milieu. These results demonstrate that MOZ/MORF KATs are a unique vulnerability of *AF10* translocation-induced leukemia in vivo.

## Dual inhibition of MOZ/MORF KATs and DOT1L KMT cooperatively induces differentiation

To elucidate the mechanisms of the antitumor effects of MOZ/MORF and DOT1L inhibition, we identified the genes directly regulated by ENL in P31/FUJ cells. ChIP-seq analysis of P31/FUJ cells using a specific anti-ENL antibody showed that ENL was localized to the *MYC*, *MYB*, *MEIS1*, and *HOXA* genes (Fig. 7a). A surrogate marker for active transcription, RNAP2 with its CTD phosphorylated at Ser5 on the heptapeptide motif (RNAP2 Ser5-P), was detected at these ENL-target genes. The average distribution of ENL and RNAP2 Ser5-P at the promoter-proximal coding regions was less prominent in P31/FUJ cells compared to that in HEK293T cells probably due to low protein expression of the components of MLL/MOZ/AEP-mediated transcriptional activation system (Supplementary Fig. 8a–c), suggesting that general transcription is relatively low in P31/FUJ cells partly due to the low presence of ENL. The expression of these ENL-target genes was suppressed by the MOZ/MORF KAT and DOT1L KMT inhibitors in a time-dependent manner (Fig. 7b). After the inhibition of MOZ/MORF KATs or DOT1L KMT, *ITGAM* (the gene coding for CD11b) expression substantially increased, indicating that myeloid differentiation was induced. Owing to the low protein expression of endogenous MOZ/MORF KATs and DOT1L (Supplementary Fig. 8c), ChIP analysis of their genomic localization using available antibodies was technically difficult; therefore, we performed ChIP-qPCR analysis of ENL (a shared component of the MOZ, AEP, and DOT1L complexes), ING4 (a MOZ complex component), Cyclin T1 (an AEP component), and RNAP2 Ser5-P on P31/FUJ cells with WM1119 treatment (Fig. 7c). WM1119 treatment reduced the presence of ENL and its cofactors at these target genes, accompanied by a reduction of the ChIP signal of RNAP2 Ser5-P. This suggests that WM1119 displaces ENL from the target promoters and impairs AEP-mediated transcriptional activation in CALM-AF10 leukemia cells.

EPZ-5676 more profoundly downregulated the expression of *MEIS1*, *HOXA5*, and *HOXA9*, compared to that of *MYC* and *MYB* (Fig. 7b). Therefore, the dependencies in MOZ/MORF KATs and DOT1L KMT in CALM-AF10-mediated gene activation seemed to slightly differ for each key oncogene.

A greater reduction in oncogene expression and more advanced differentiation were induced at lower concentrations when the two drugs were simultaneously added to the culture (Fig. 8a–c). A similar cooperative effect was seen on another leukemia cell line carrying CALM-AF10 fusion, termed KP-Mo-TS[40] (Supplementary Fig. 9a). In contrast, the proliferation of normal human CD34 + cells was unaffected under the conditions where that of P31/FUJ cells was severely impaired (Fig. 8d, e), suggesting that the combined inhibition of MOZ/MORF KATs and DOT1L KMT selectively induces differentiation of CALM-AF10 leukemia cells. Next, we examined the effects of the dual inhibition of MOZ/MORF KATs and DOT1L KMT in vivo. Administration of WM1119 (35 mg/kg) and EPZ-5676 (20 mg/kg) three times per day for 2 weeks substantially reduced the leukemia burden compared to the single-drug treatment (Fig. 8f and Supplementary Fig. 9b). Thus, the combination therapy with MOZ/MORF KATs and DOT1L KMT inhibitors could improve therapeutic outcomes in *AF10* translocation-induced leukemia.

## Discussion

In this study, we showed that AF10 fusions exert their leukemic potential via the MOZ-ENL-DOT1L axis. The ENL protein in the AF10 fusion complexes requires an intact YEATS domain that binds to acetylated histone H3 lysine 18/27 (H3K18/27ac)[31,32] and MOZ/MORF KATs[7], and an intact AHD domain that binds to DOT1L and AF4 family proteins[8]. XPO1 is reportedly associated with the NES in CALM[33] and fuses with AF10 in human leukemia[41]. The current paradigm states that XPO1 serves as a recruiting factor that binds to the *HOXA* loci and recruits AF10 fusion complexes[33]. However, our results suggest an alternative mechanism in which ENL binds to the *HOXA* loci via MOZ/MORF KATs, recruiting AF10 fusion/DOT1L complexes (Fig. 9). We have previously shown that MOZ targets CpG-rich promoters via association with MLL and RNAP2 complexes[7]. Here, we showed that ENL and DOT1L were localized at the *HOXA* loci in a MOZ-dependent manner in the absence of the CALM-AF10 fusion (Fig. 3e). Thus, the

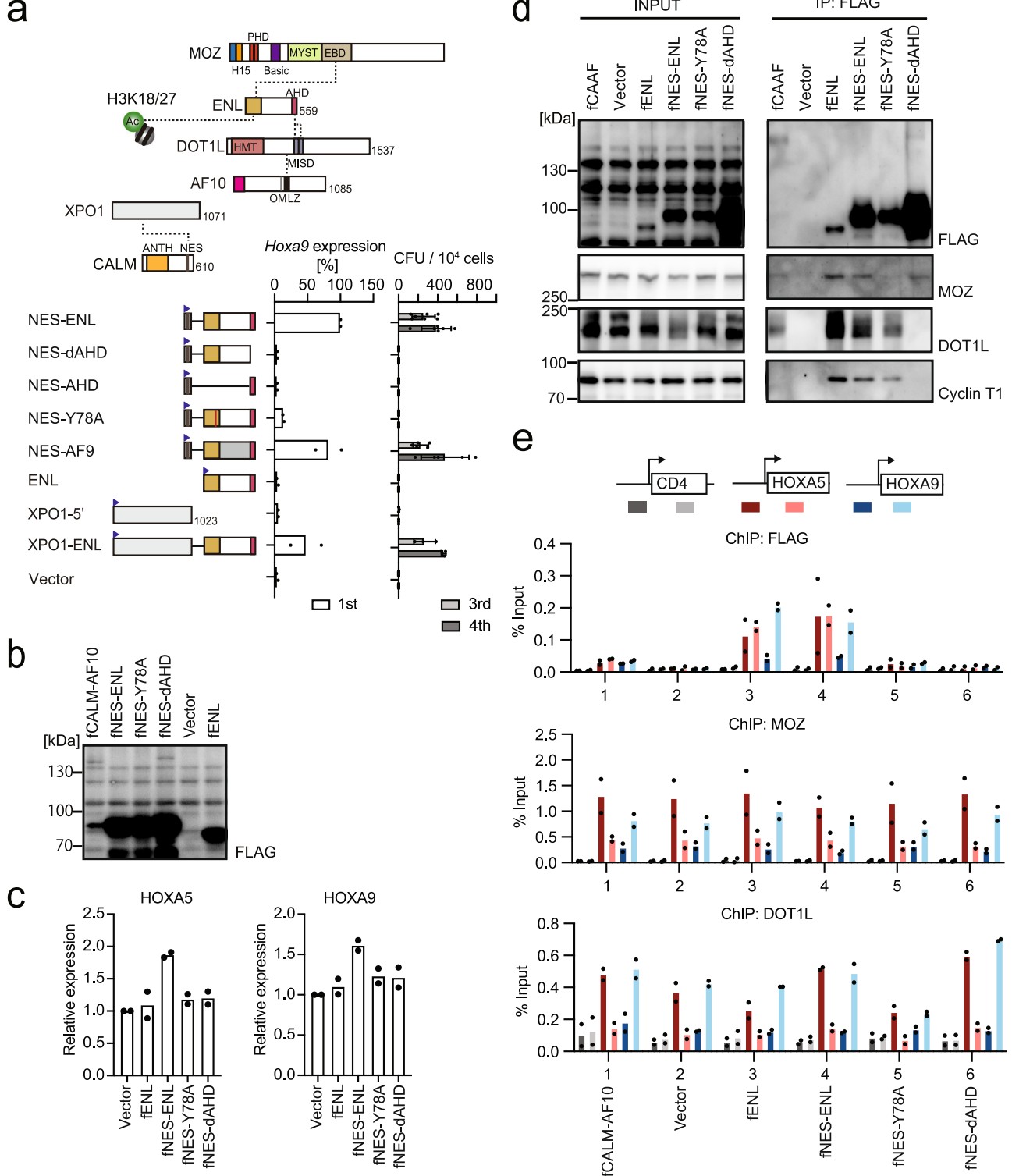

**Fig. 3 | Structural requirements of NES-ENL in overactivation of *HOX* genes.**
**a** Structures required for NES-ENL-mediated transformation. Various ENL constructs were examined for the transformation of myeloid progenitors as in Fig. 1a.
**b** Protein expression of various ENL constructs in stably transduced HEK293T cells. FLAG-tagged (denoted as f) gene constructs were lentivirally transduced to HEK293T cells and selected for puromycin resistance. WB of the whole-cell extracts was performed using anti-FLAG antibodies. **c** *HOX* gene expression in HEK293T cells stably expressing ENL mutants. RT-qPCR was performed on the cells presented in **c** (mean of two technical replicates). Multiple comparisons were performed using a one-way ANOVA. **d** Protein interaction of NES-ENL in the chromatin fraction. IP-WB of the chromatin fraction of HEK293T cells stably expressing various FLAG-tagged ENL constructs was performed using specific antibodies for the indicated proteins. **e** Localization of the various ENL constructs and CALM-AF10. ChIP-qPCR for the FLAG-tagged proteins, MOZ, and DOT1L were performed as in Fig. 2d. (mean of two technical replicates). Source data are provided as a Source Data file.

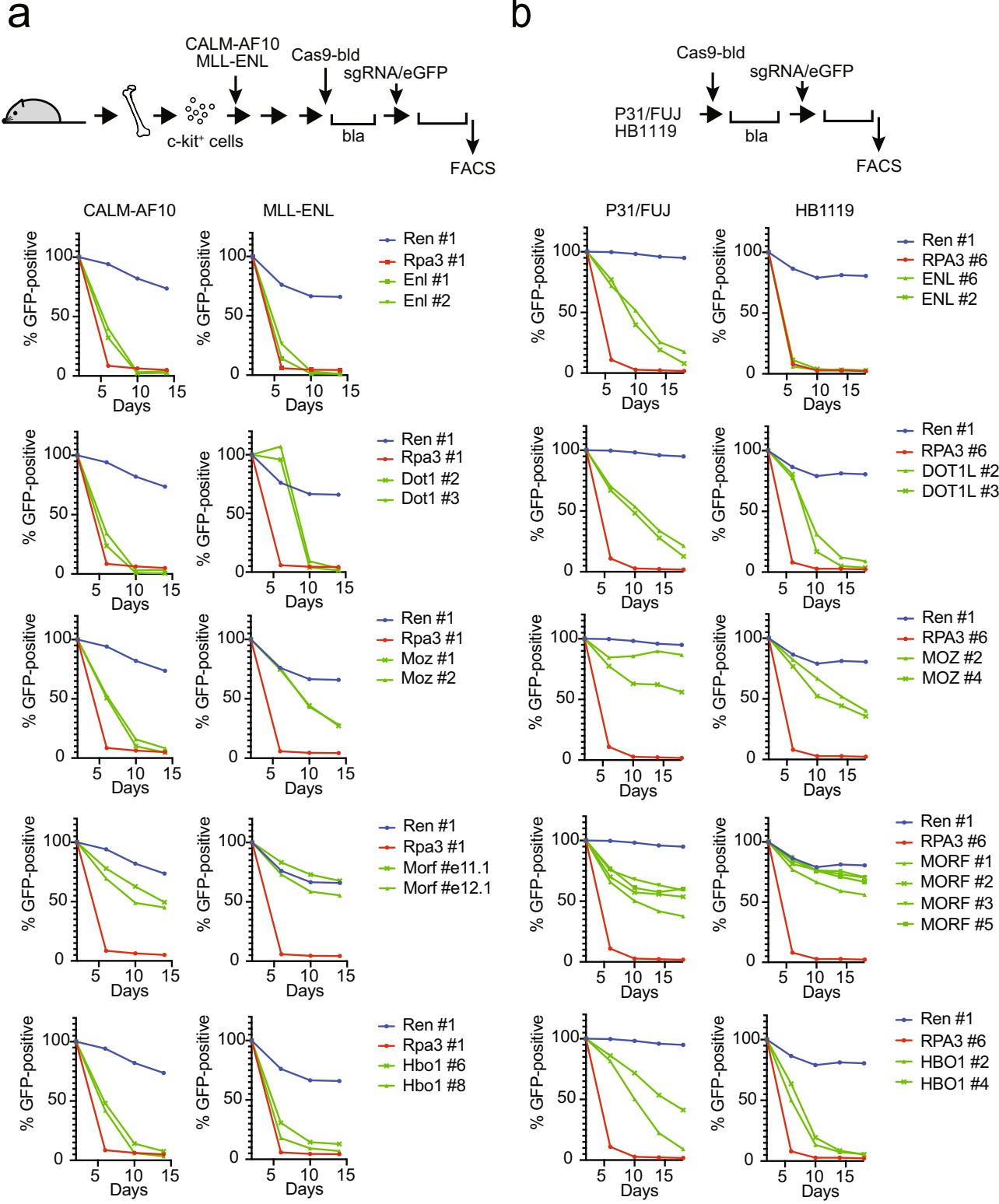

**Fig. 4 | ENL co-factors and MYST family KATs determine survival in leukemia cells. a** Requirement of DOT1L, ENL, MOZ, MORF, and HBO1 for myeloid progenitors immortalized by CALM-AF10 or MLL-ENL. sgRNA competition assays of immortalized myeloid progenitors were performed. The ratio of GFP-positive cells co-expressing sgRNA was measured using flow cytometry. sgRNA for Renilla luciferase (Ren), which does not affect proliferation, was used as a negative control.

sgRNA for Rpa3, which impairs proliferation, was used as a positive control. **b** Requirement of DOT1L, ENL, MOZ/MORF, and HBO1 for human leukemia cell lines carrying CALM-AF10 (P31/FUJ) or MLL-ENL (HB1119). sgRNA competition assays were performed in P31/FUJ and HB1119 cells as in a. Source data are provided as a Source Data file.

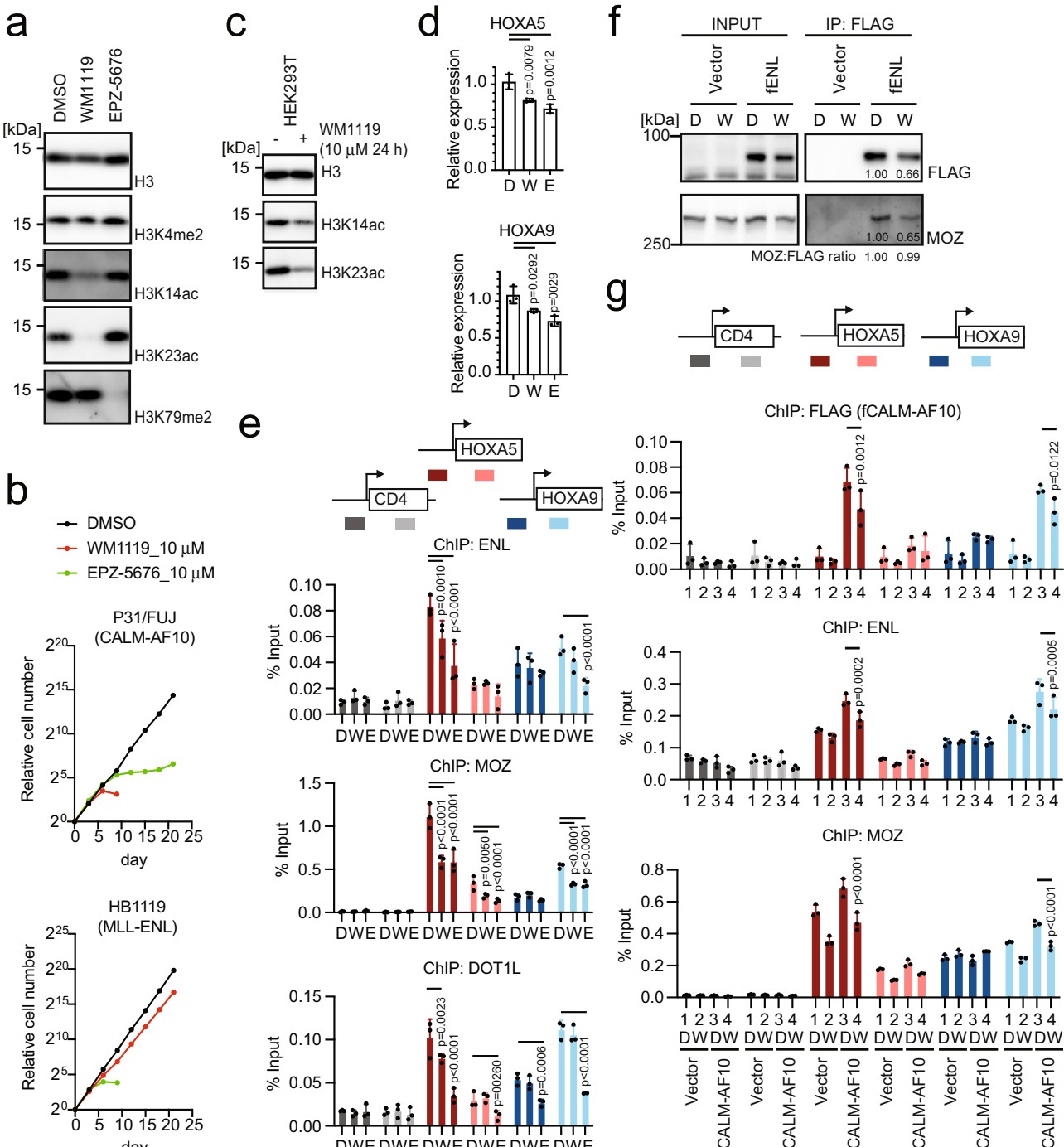

**Fig. 5 | MOZ/MORF-mediated acetylation is required for the localization of the MOZ/ENL complex and CALM-AF10 at the target chromatin. a** Histone modifications in CALM-AF10 leukemia cells after treatment with MOZ/MORF KAT (WM1119) and DOT1L KMT (EPZ-5676) inhibitors. WB of the P31/FUJ cells treated with 10 μM of WM1119 or EPZ-5676 for 6 days using antibodies specific to each histone modification. **b** Leukemia cell proliferation in the presence of WM1119 and EPZ-5676. P31/FUJ cell proliferation in the presence of 10 μM WM1119 or EPZ-5676 was monitored for 21 days along with the vehicle control (DMSO). **c** Histone modifications in HEK293T cells after treatment with WM1119. WB of the HEK293T cells treated with 10 μM of WM1119 for 24 h using antibodies specific to each histone modification. **d** *HOXA* gene expression in HEK293T cells after treatment with WM1119 and EPZ-5676. RT-qPCR of the HEK293T cells treated with 10 μM of WM1119 (W), EPZ-5676 (E), or DMSO control (D) for 5 days (mean ± SD of technical replicates, *n* = 3). Multiple comparisons were performed using a one-way

ANOVA. **e** Localization of ENL, MOZ, and DOT1L after treatment with WM1119 and EPZ-5676. ChIP-qPCR of HEK293T cells treated with 10 μM of WM1119 or EPZ-5676 for 5 days was performed as in Fig. 2d (mean ± SD of technical replicates, *n* = 3). Multiple comparisons were made using a two-way ANOVA. **f** Effects of WM1119 on protein interactions in ENL. IP-WB of the chromatin fraction of HEK293T cells stably expressing various FLAG-tagged ENL constructs with (W: WM1119) or without (D: DMSO control) the treatment with WM1119 at 10 μM for 5 days was performed. The co-precipitates were analyzed using anti-FLAG and MOZ antibodies. The relative band intensity was quantified using the Image J software. **g** Localization of CALM-AF10 after treatment with WM1119. HEK293T cells stably expressing FLAG-tagged CALM-AF10 (fCALM-AF10) were treated with 10 μM WM1119 (W) or the DMSO control (D) for 5 days and subjected to ChIP-qPCR analysis as in Fig. 2d (mean ± SD of technical replicates, *n* = 3). Multiple comparisons were performed using a two-way ANOVA. Source data are provided as a Source Data file.

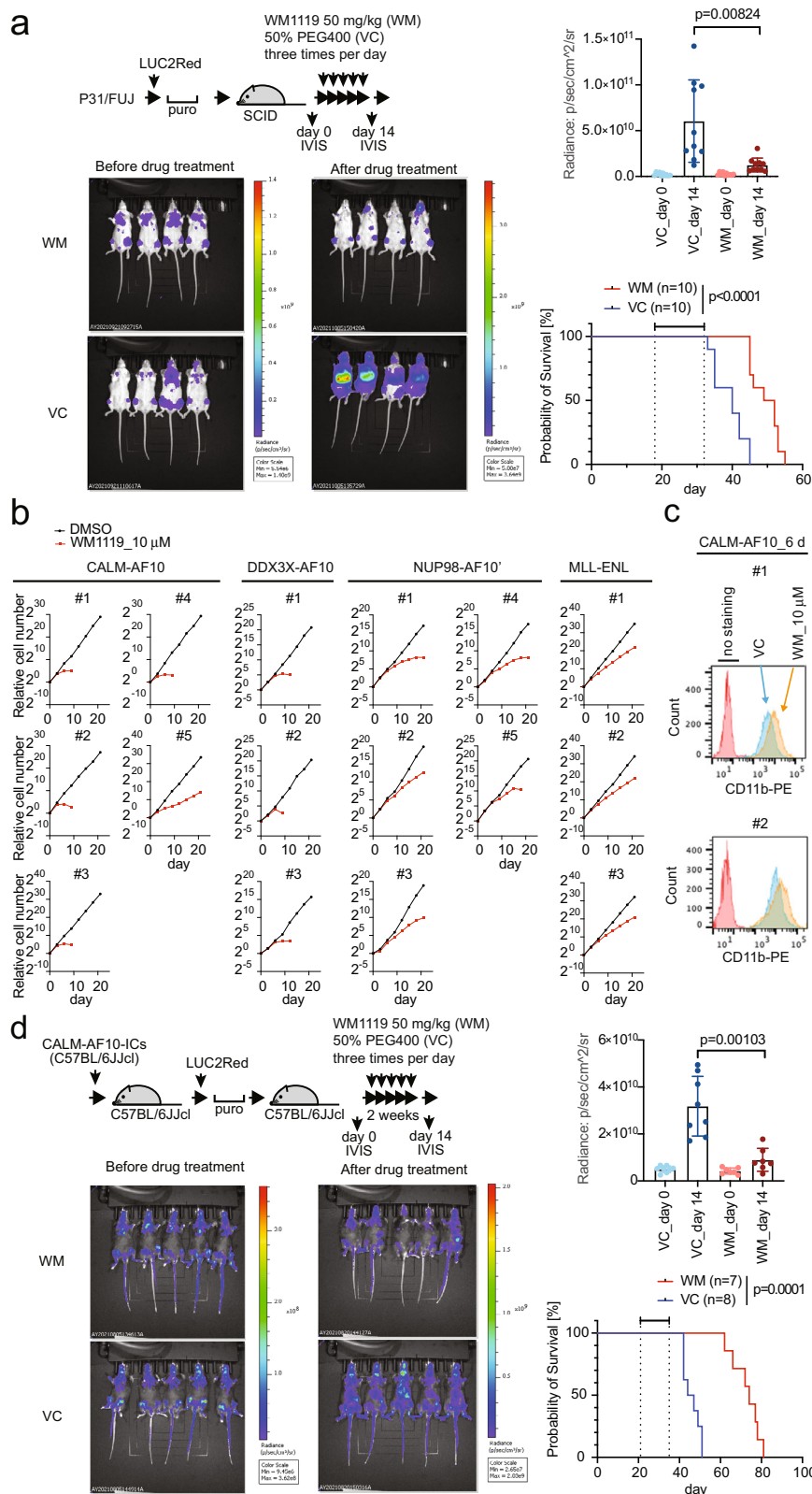

NES-XPO1 interaction is unlikely to be a decisive targeting mechanism but is rather required for the overactivation of *HOXA* genes. MLL recruits CBP/p300 KATs to provide H3K18/27ac marks on which the AEP complex is formed[7]. In *MLL* translocation-induced leukemias, constitutive recruitment of AEP to CpG-rich promoters by MLL fusions drives leukemic transformation, as an artificial fusion protein of the targeting module and the AHD domain, which lacks the YEATS domain,

was able to transform HPCs (see MTM-AHD in Fig. 1a). In contrast, CALM-AF10 requires an intact YEATS domain of ENL that confers targeting ability to MOZ/MORF complexes, suggesting that emerging YEATS domain inhibitors may be highly effective against *AF10* translocation-induced leukemia[42–44]. Our results support a model where ENL plays a critical role as a targeting factor for MOZ/MORF-occupied chromatin (Fig. 9). A portion of NUP98 proteins has been

**Fig. 6 | Pharmacological inhibition using the MOZ/MORF KAT inhibitor suppresses CALM-AF10-mediated leukemia in vivo. a** Leukemia burden before and after MOZ/MORF KAT inhibitor treatment. P31/FUJ human leukemia cells expressing LUC2Red were transplanted into sublethally irradiated SCID mice and intraperitoneally injected with WM1119 50 mg/kg (WM) or the 50% PEG400 vehicle control (VC) three times a day for 2 weeks, as shown in the top schema. Representative images of luciferase-mediated bioluminescence reflecting leukemia cell abundance are shown with scale bars. Quantification of leukemia burden by region-of-interest analysis of the IVIS images (mean ± SD of biological replicates, $n = 10$). Welch's *t*-test (two-sided) was performed on the indicated two-group comparison. Survival of transplanted recipient mice with or without MOZ/MORF KAT inhibitor treatment (the duration of drug administration is indicated in the graph). A log-rank (Mantel–Cox) test was performed. **b** Proliferation of various AF10 fusion-ICs in the presence of MOZ/MORF KAT inhibitor (10 μM WM1119) was monitored for 21 days along with the vehicle control (DMSO). **c** Differentiation induced by MOZ/MORF KAT inhibitors. Flow cytometry analysis for the CD11b antigen was performed on the CALM-AF10-ICs treated with 10μM of WM1119 for 6 days. **d** Leukemia burden before and after MOZ/MORF KAT inhibitor treatment. Murine CALM-AF10 leukemia cells expressing LUC2Red were transplanted into sub-lethally irradiated syngeneic C57BL/6JJcl mice, treated with WM1119 as in **a**. Representative images of luciferase-mediated bioluminescence are shown as in a. Quantification of leukemia burden by region-of-interest analysis of the IVIS images (mean ± SD of biological replicates, $n \geq 7$). Welch's *t*-test (two-sided) was performed on the indicated two-group comparison. Survival of transplant recipient mice with or without MOZ/MORF KAT inhibitor treatment (the duration of drug administration is indicated in the graph). A log-rank (Mantel–Cox) test was performed. Source data are provided as a Source Data file.

shown to localize in a nucleoplasmic fraction away from the nuclear pore complex and associates with an active transcriptional factory in eukaryotes[45–47], likely as a complex with XPO1 and NES-containing proteins[48]. Thus, the NES-XPO1-NUP98 interaction may play an active role in transcriptional activation by sequestering the *HOXA* loci in the proximity of the active transcriptional factories.

We propose that the inhibition of MOZ/MORF KATs can be used as a therapeutic strategy for *AF10* translocation-induced leukemia. MOZ/MORF-mediated acetylation activity appears to be uniquely required for AF10-mediated leukemogenesis (Figs. 5, 6). We found that the presence of the MOZ/ENL complex on chromatin was decreased upon MOZ/MORF KAT inhibition (Fig. 5e, g). This decrease of the MOZ/ENL complex induces the dissociation of CALM-AF10 from the target chromatin. Mechanistically, the removal of H3K14/23ac may attenuate the chromatin binding of the MOZ/ENL complex because it contains a reader module for H3K14/23ac[49] (Fig. 9).

WM1119 has shown negligible adverse side effects in vivo[36], and exhibited some antitumor effects toward *MLL* translocation-induced leukemia ex vivo[50]. In this study, we demonstrated excellent antitumor effects of WM1119 in vivo in murine models of *AF10* translocation-induced leukemia. Importantly, WM1119 was much more effective in *AF10* translocation-induced leukemia than in *MLL* translocation-induced leukemia likely due to its direct effects on chromatin targeting by AF10 fusion proteins. ENL-target genes had locus-specific differential dependencies on DOT1L KMT, where DOT1L had a greater influence on *MEIS1*, *HOXA5*, and *HOXA9* compared to *MYC* and *MYB* in P31/FUJ cells (Fig. 7b). Simultaneous inhibition of MOZ/MORF KATs and DOT1L KMT induced marked reductions in the expression of these key oncogenes and exerted an antitumor effect at lower concentrations. Thus, the MOZ/MORF KAT inhibitor and its combination with DOT1L KMT inhibitors may provide novel therapeutic strategies for this malignant subtype of leukemia.

## Methods

### Vector constructs
For protein expression vectors, cDNAs obtained from Kazusa Genome Technologies Inc[51]. or DNA fragments synthesized by DNA synthesis services were cloned into the pMSCV (for retrovirus production), pCDH (for lentivirus production), or pCMV5 (for transient expression) vector using restriction enzyme digestion and DNA ligation. The MSCV-neo MLL-AF10 vectors have been previously described[8]. sgRNA-expression vectors were constructed using the pLKO5.sgRNA.EFS.GFP vector[52]. The target sequences of sgRNA are listed in Supplementary Tables 1, 2.

### Cell lines
In the earlier phase of the study, we used HEK293T cells donated by Michael Cleary, which were authenticated by the JCRB Cell Bank in 2019. Later, we used the HEK293T cells purchased from ATCC.

HEK293T cells deficient of *MOZ* and *DOT1L* were generated by transfection of pX335-U6-Chimeric_BB-CBh-hSpCas9n(D10A)-based gene editing vectors[53] carrying two different sgRNAs for each gene and subcloning. Colonies derived from a single-cell clone were expanded and examined for protein expression by WB. The sequences for sgRNA are listed in Supplementary Table 3. Cells were cultured in Dulbecco's modified Eagle's medium (DMEM), supplemented with 10% fetal bovine serum (FBS) and penicillin-streptomycin (PS). The platinum-E (PLAT-E) ecotropic virus-packaging cell line—a gift from Toshio Kitamura[54]—was cultured in DMEM supplemented with 10% FBS, puromycin, blasticidin, and PS. The human leukemia cell lines P31/FUJ (purchased from JCRB), HB1119 (a gift from Michael Cleary)[55], U937 (purchased from JCRB), MonoMac6 (purchased from DSMZ), and K562 (a gift from Michael Cleary)[55] were cultured in RPMI 1640 medium supplemented with 10% FBS and PS. KP-Mo-TS cells (a gift from Issay Kitabayashi)[40] were cultured in RPMI 1640 medium supplemented with 20% FBS and PS. MV4-11 cells (purchased from ATCC) were cultured in Iscove's Modified Dulbecco's medium (IMDM) supplemented with 10% FBS and PS. OCI-AML3 cells (purchased from DSMZ) were cultured in α-Minimum Essential Media (MEM) with 20% FBS and PS. Murine myeloid progenitors immortalized by various transgenes were cultured in RPMI 1640 medium supplemented with 10% FBS and PS, containing murine stem cell factors (SCF), interleukin-3 (IL-3), and granulocyte-macrophage colony-stimulating factor (GM-CSF; 1 ng/mL of each). Cells were cultured in an incubator at 37 °C and 5% $CO_2$ and routinely tested for mycoplasma using a MycoAlert Mycoplasma Detection Kit (Lonza).

### Western blotting (WB)
Proteins were separated electrophoretically on an SDS-PAGE gel and transferred onto nitrocellulose sheets using a mini transblot cell (Bio-Rad). The nitrocellulose sheets were blocked with 5% skim milk in PBS-T (phosphate-buffered saline containing 0.1% Tween 20) for 1 h, rinsed twice with PBS-T, and incubated with primary antibodies (listed in Supplementary Table 6) in a 5% skim milk and PBS-T suspension overnight. The blots were then washed twice with PBS-T and incubated with peroxidase-conjugated secondary antibodies (listed in Supplementary Table 6) for 3 h in a suspension of 5% skim milk in PBS-T. Chemiluminescence was performed using an ECL chemiluminescence reagent (GE Healthcare). The antibodies used in this study are listed in Supplementary Table 6. The uncropped scans of the blots are included in the Source Data file.

### Virus production
Ecotropic retroviruses were produced using PLAT-E packaging cells (a gift from Toshio Kitamura)[54]. Lentiviruses were produced in HEK293T cells using the pMDLg/pRRE, pRSV-rev, and pMD2.G vectors, all of which were gifts from Didier Trono[56]. Virus-containing media were harvested 24–48 h after transfection and were used for viral transduction.

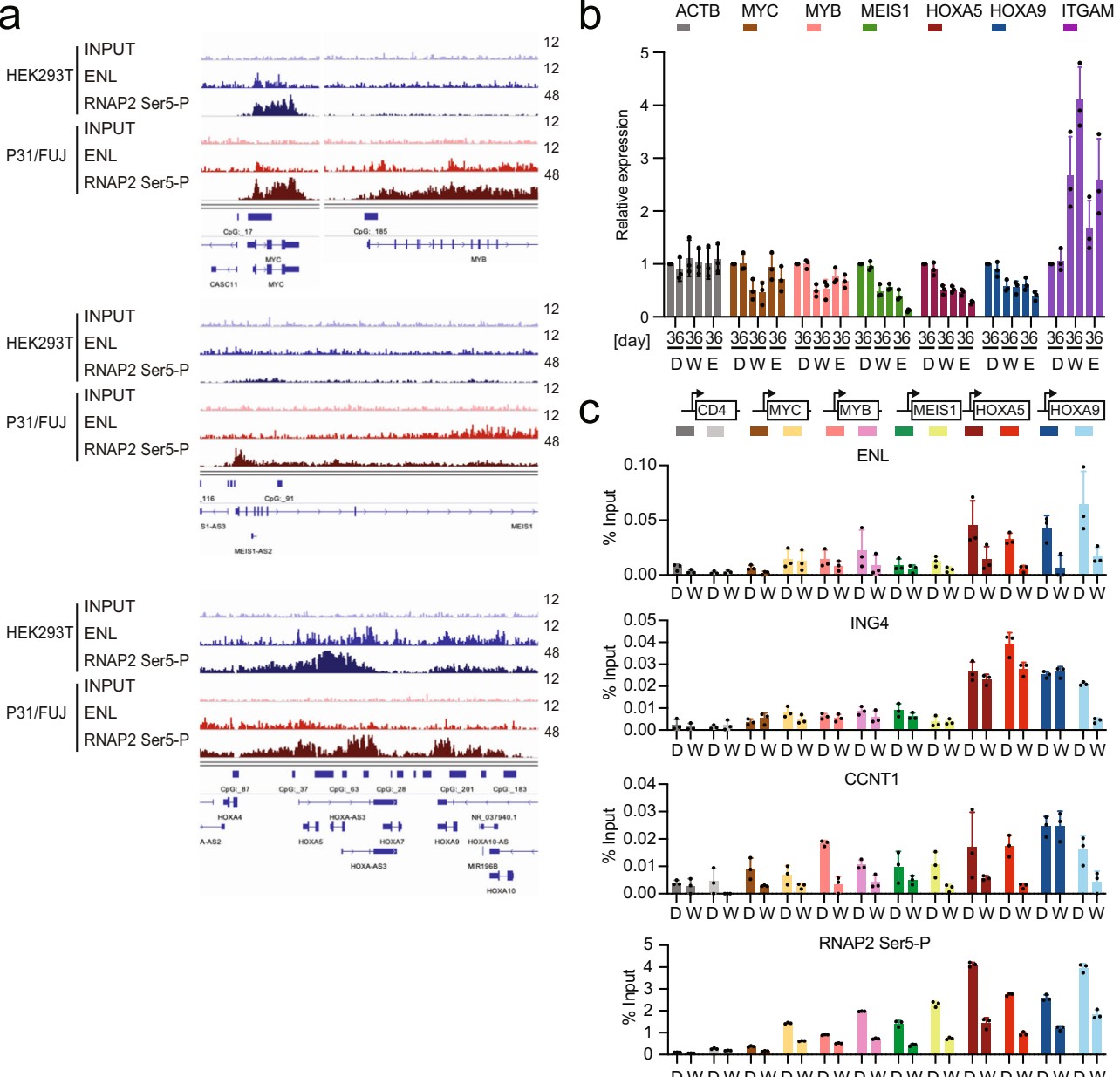

**Fig. 7 | Inhibition of MOZ/MORF KATs reduces ENL-mediated oncogene activation in CALM-AF10 leukemia cells. a** Genomic localization of ENL. ChIP-seq for ENL was performed on P31/FUJ and HEK293T cells. Data for RNA Polymerase II with its CTD phosphorylated at Ser 5 of the heptapeptide repeats (RNAP2-Ser5-P) are shown as a comparison. **b** Gene expression after MOZ/MORF KAT or DOT1L KMT inhibition. RT-qPCR was performed on the P31/FUJ cells treated with 10 μM WM1119 (W), 10 μM EPZ-5676 (E), or DMSO control (D) for 3 or 6 days (mean ± SD of technical replicates, *n* = 3). **c** Effects of MOZ/MORF KAT inhibitor on the localization of ENL, ING4, Cyclin T1, and RNAP2-Ser5-P at target chromatin. ChIP-qPCR was performed on P31/FUJ cells treated with WM1119 (W) (1 μM for 24 h) or DMSO control (D) (mean ± SD of technical replicates, *n* = 3) using specific antibodies for the indicated proteins. The positions of the qPCR probe are set at 0.5 kb upstream and 1.5 kb downstream of TSS as indicated on the top. Source data are provided as a Source Data file.

## Myeloid progenitor transformation assay

The myeloid progenitor transformation assay was performed as previously described[24]. BM cells were harvested from the femurs and tibiae of 5-week-old female C57BL/6 J mice. c-Kit+ cells were enriched using magnetic beads conjugated with an anti-c-Kit antibody (Miltenyi Biotec), transduced with a recombinant retrovirus by spinoculation, and plated (4 × 10⁴ cells/sample) in a methylcellulose medium (IMDM, 20% FBS, 1.6% methylcellulose, and 100 μM β-mercaptoethanol) containing murine SCF, IL-3, and GM-CSF (10 ng/mL each). During the first culture passage, G418 (1 mg/mL) was added to the culture medium to select the transduced cells. *Hoxa9* expression was quantified using qRT-PCR after the first passage. Cells were then re-plated once every

5 days with a fresh medium. Colony-forming units were quantified per 10⁴ plated cells for each passage.

## In vivo leukemogenesis assay

The in vivo leukemogenesis assay was performed as previously described[57]. Cells positive for c-Kit (2 × 10⁵), prepared from mouse femurs and tibiae, were transduced with retroviruses by spinoculation and intravenously transplanted into sublethally irradiated (6 Gy) 7–8-week-old female C57BL/6 JJcl (C57BL/6J) mice. Moribund mice were euthanized, and the cells isolated from the BM were freeze-stocked and later subjected to secondary transplantation. For secondary leukemia, leukemia cells (2 × 10⁵) were transplanted in the same manner as

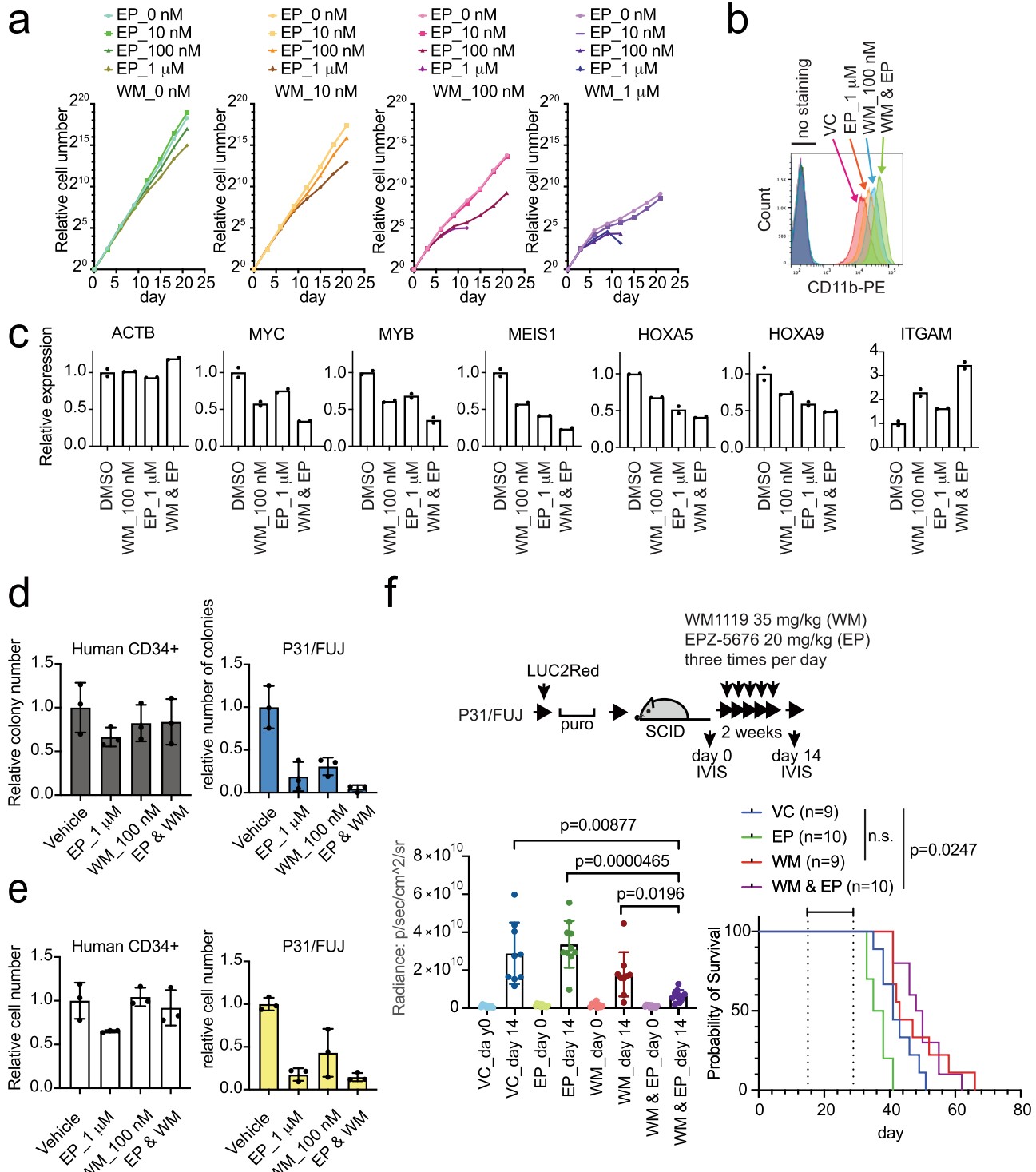

**Fig. 8 | Dual inhibition of MOZ/MORF KATs and DOT1L KMT cooperatively induces differentiation of CALM-AF10 leukemia cells. a** Effects on the proliferation of P31/FUJ cells by dual inhibition of MOZ/MORF KATs and DOT1L KMT. Proliferation of P31/FUJ cells in the presence of WM1119 and EPZ-5676 at the indicated concentrations was monitored for 21 days. **b** Differentiation of P31/FUJ cells after combinatorial inhibition of MOZ/MORF KAT and DOT1L KMT. Flow cytometry analysis for the CD11b antigen was performed on the P31/FUJ cells treated with various combinations of WM1119 and EPZ-5676 for 6 days. **c** Effects of combinatorial inhibition of MOZ/MORF KAT and DOT1L KMT on gene expression. RT-qPCR was performed on the P31/FUJ cells treated with only 100 nM WM1119 (WM), only 1 μM EPZ-5676 (EP), or a combination of both (WM & EP) (mean of two technical replicates). **d** Effects on the colony formation of human CD34 + cells and P31/FUJ cells by dual inhibition of MOZ/MORF KATs and DOT1L KMT (mean ± SD of technical replicates, $n = 3$). **e** Effects on the cell numbers of Human CD34 + cells and P31/FUJ cells by dual inhibition of MOZ/MORF KATs and DOT1L KMT (mean ± SD of three technical replicates). **f** Leukemia burden before and after the combination therapy using MOZ/MORF KATs and DOT1L KMT inhibitors. P31/FUJ human leukemia cells expressing LUC2Red were transplanted into sublethally irradiated SCID mice and subsequently administered with 35 mg/kg of WM1119 and 20 mg/kg of EPZ-5676 three times a day (i.p.) for 2 weeks (mean ± SD of biological replicates, $n \geq 9$). Welch's $t$-test (two-sided) was performed on the indicated two-group comparison. Survival of transplanted recipient mice with or without MOZ/MORF KAT inhibitor treatment (the duration of drug administration is indicated in the graph). A log-rank (Mantel–Cox) test was performed. n.s.: $P > 0.05$. Source data are provided as a Source Data file.

**Fig. 9 | MOZ-dependent recruitment of AF10 fusion complex leads to gene activation.** A model of CALM-AF10-mediated gene activation is dipcted in a cartoon.

the primary transplantation. The study protocol was approved by the Institutional Animal Care and Use Committee of the National Cancer Center.

### RT-qPCR

Total RNA was isolated using an RNeasy Mini Kit (Qiagen) and reverse-transcribed using a Superscript III First Strand cDNA Synthesis System (Thermo Fisher Scientific) with oligo (dT) primers. Gene expression was analyzed by qPCR using TaqMan probes (Thermo Fisher Scientific). Relative expression levels were normalized to those of *GAPDH/Gapdh* or *ACTB* and determined using a standard curve and

relative quantification method, as per the manufacturer's instructions (Thermo Fisher Scientific). The commercially available PCR probe sets used in this study are listed in Supplementary Table 6.

### Fractionation-assisted chromatin immunoprecipitation (fanChIP)

Chromatin fractions from HEK293T and P31/FUJ cells were prepared using the fanChIP method[34]. Cells were suspended in CSK buffer [100 mM NaCl, 10 mM PIPES (pH 6.8), 3 mM MgCl$_2$, 1 mM EGTA, 0.3 M sucrose, 0.5% Triton X-100, 5 mM sodium butyrate, 0.5 mM DTT, and protease inhibitor cocktail] and centrifuged ($400 \times g$ for 5 min at 4 °C) to remove the soluble fraction. The pellet was resuspended in MNase buffer [50 mM Tris-HCl (pH 7.5), 4 mM MgCl$_2$, 1 mM CaCl$_2$, 0.3 M sucrose, 5 mM sodium butyrate, 0.5 mM DTT, and protease inhibitor cocktail] and treated with MNase at 37 °C for 3–6 min to obtain oli-gonucleosomes. The MNase reaction was then stopped by adding EDTA (pH 8.0) to a final concentration of 20 mM. Equal amounts of lysis buffer [250 mM NaCl, 20 mM sodium phosphate (pH 7.0), 30 mM sodium pyrophosphate, 5 mM EDTA, 10 mM NaF, 0.1% NP-40, 10% glycerol, 1 mM DTT, and EDTA-free protease inhibitor cocktail] were added to increase solubility. The chromatin fraction was cleared by centrifugation ($17,700 \times g$ for 5 min at 4 °C) and subjected to IP with specific antibodies and protein-G magnetic microbeads (Invitrogen) or with anti-FLAG M2 antibody-conjugated beads (Supplementary Table 6). The immunoprecipitates were then washed five times with washing buffer (1:1 mixture of lysis buffer and MNase buffer with 20 mM EDTA) and eluted in elution buffer (1% SDS and 50 mM NaHCO$_3$). The eluted material was analyzed by WB, qPCR, and deep sequencing.

### RNA-Seq

Total RNA was extracted from cells prepared using RNeasy Kit (Qiagen), and its quantity and purity were analyzed using a Bioanalyzer (Agilent Technologies). Deep sequencing was performed using a Tru-Seq Stranded mRNA Library Prep Kit (Illumina) and HiSeq2500 (Illumina) with 51-bp single-end reads at the core facility of Hiroshima University. Sequenced reads were mapped to the mouse genome assembly mm9 using STAR[58], and read counts were analyzed using 'featureCounts'[59]. Differentially expressed genes (DEGs) (False Discovery Rate (FDR) < 0.05) were obtained using TCC R package[60,61]. The gene set of DEGs with < 0.01 FDR was analyzed using Metascape[62] and GSEA[63] for pathway enrichment analyses. Hierarchical clustering and principal component analysis were performed using R. DEGs were visualized using the ComplexHeatmap R package[64].

### ChIP-qPCR and ChIP-seq

The material eluted by fanChIP was extracted using phenol/chloro-form/isoamyl alcohol. DNA was precipitated with glycogen, dissolved in TE buffer, and analyzed by qPCR (ChIP-qPCR) or deep sequencing (ChIP-seq). The qPCR probe/primer sequences are listed in Supplementary Table 4. For deep sequencing, the DNA was further fragmented (-150 bp) using a Covaris M220 DNA shearing system (M&M Instruments Inc.). Deep sequencing was then performed using a Tru-Seq ChIP Sample Prep Kit (Illumina) at the core facilities of Hiroshima University and the University of Tokyo. Data were visualized using Integrative Genome Viewer (The Broad Institute). Raw reads in the FASTQ format were trimmed using Cutadapt and aligned to the reference genome hg19 using BWA[65,66]. Accession numbers and sample IDs are listed in Supplemental Table 5.

### sgRNA competition assay

*Cas9* was introduced via lentiviral transduction using a pKLV2-EF1a-Cas9Bsd-W vector[67]. Cas9-expressing stable lines were established using blasticidin (10−30 μg/mL) selection. The sgRNA sequences, listed in Supplemental Tables 1, 2, were either designed using the CCTop

design tool[68] or adopted from previous reports[11,69]. The targeted sgRNA was co-expressed with GFP via lentiviral transduction using the pLKO5.sgRNA.EFS.GFP vector[52]. Percentages of GFP+ cells were determined by FACS analysis 2 days post-sgRNA transduction and then measured every 4 days.

## FACS analysis

One million cells were incubated with PE-conjugated anti-CD11b antibody (Supplementary Table 6) for 1 h on ice, and washed with 1 mL of staining media (PBS containing 3% FBS). Then, the cells were centrifuged, resuspended in fresh staining media, and analyzed with FACS Melody (BD Bioscience). Alive cells were gated as in Supplementary Fig. 10.

## Ex vivo culture of human CD34 + cells

Human hematopoietic stem cell−CD34 + cells the from fetal liver (purchased from Cell Applications, INC.) were pre-cultured in Hematopoietic Stem Cell Culture Medium (Cell Applications, INC.) for 4 days and plated in methylcellulose-based medium (MethoCult™ H4435 Enriched, STEMCELL Technologies) in the presence of designated drugs for 6 days. For comparison, P31/FUJ cells were also plated in the same conditions for 15 days.

## Drug administration and monitoring by IVIS

The *LUC2* luciferase gene in pGL4.10 (AY737822) was modified by introducing a Ser284Thr mutation[70]. The modified *LUC2* gene (*LUC2Red*) was fused to the P2A self-cleaving peptide sequence and the puromycin resistance gene; then, it was cloned into a pCDH vector (System Biosciences, LLC). Human P31/FUJ leukemia cells and murine CALM-AF10 leukemia cells (CAAF-bm0317) were transduced using lentiviral gene transfer and maintained in puromycin. P31/FUJ-LUC2Red cells were transplanted into sublethally irradiated (2.5 Gy) SCID (C.B-17/Icr-scid/scidJcl) mice (7-week-old females). CAAF-bm0317-LUC2Red cells were transplanted into sublethally irradiated (6 Gy) C57BL/6JJcl mice (7−8-week-old females). Leukemia burden was monitored using an in vivo imaging system (IVIS)(Lumina LT, Perkin Elmer) and D-luciferin (VivoGlo Luciferin, in vivo grade, Promega) as per the manufacturer's instruction.

## Statistics and reproducibility

Statistical analysis was performed using GraphPad Prism 8 and Microsoft Excel software. Data are presented as mean ± standard deviation (SD). Multiple comparisons were performed using one-way or two-way ANOVA; all statistical tests were two-sided. Welch's *T*-test was performed on two-group comparisons. Mouse transplantation experiments were analyzed by a log-rank (Mantel-Cox) test. Statistical significance was set at $P \leq 0.05$. All the experiments with WB analysis were independently performed at least twice and confirmed their reproducibility.

## Study approval

All the animal experimental protocols were approved by the Institutional Animal Care and Use Committee of the National Cancer Center (Tokyo, Japan).

## Reporting summary

Further information on research design is available in the Nature Portfolio Reporting Summary linked to this article.

## Data availability

All data supporting the findings of this study are available within the article, in the supplementary information, and in the source data. ChIP-seq and RNA-seq data have been deposited at the DDBJ (DNA Data Bank of Japan) Sequence Read Archive as fastq files [https://ddbj.nig.ac.jp/public/ddbj_database/dra/fastq/] and as WIG files [https://ddbj.nig.ac.jp/public/ddbj_database/gea/experiment/E-GEAD-000/] under

the accession numbers and sample IDs listed in Supplementary Table 5. Further information and requests for resources and reagents should be directed to and will be fulfilled by Akihiko Yokoyama (ayokoyam@ncc-tmc.jp). Source data are provided with this paper.

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

## Acknowledgements

We thank Hagumu Sato, Ikuko Yokoyama, Kanae Ito, Megumi Enmi, Makiko Okuda, Yuzo Sato, Megumi Nakamura, Etsuko Kanai, Ayako Yokoyama, Satoshi Takahashi, Ryo Miyamoto, and Hiroshi Okuda for technical assistance. We also thank the members of the Shonai Regional Industry Promotion Center for their administrative support. This work was supported by the Japan Society for the Promotion of Science (JSPS) KAKENHI grants (19H03694 to A.Y.; 22H03109 to A.K., and A.Y.; 22KK0119 to Y.K., A.K., and A.Y.; 17H01567 and 20H05699 to T.M.). This work was also supported in part by research funds from the Yamagata prefectural government and the city of Tsuruoka.

## Author contributions

Y.K. and A.Y. conducted the experiments; A.K. and T.I. performed deep sequencing; A.K. analyzed the deep sequencing data; T.M. provided the CRISPR-mediated screening data. A.Y. conceived the project; A.Y. supervised the project; A.Y. wrote the paper.

## Competing interests

A.Y. received a research grant from Sumitomo Pharma Co. Ltd. The remaining authors declare no competing interests.
