## [Peer Review File · Nature Communications]

REVIEWER COMMENTS

Reviewer #1 (Remarks to the Author):

The manuscript by Komata et al describes a role of MOZ and ENL in AF10 fusion proteins-mediated gene expression in leukemia. Using an ex vivo myeloid progenitor transformation assay and artificial gene constructs encoding various combined essential protein domains, the authors show that ENL is essential in AF10-fusions-mediated leukemic transformation. ENL occupies promoters of target genes that were also occupied by the MOZ, DOT1L, AEP, and HBO1 complexes, likely forming a positive feedback regulatory loop. Genetic ablation or pharmacological inhibition of the MOZ/MORF KATs suppresses CALM-AF10-mediated leukemia growth both in vitro and in vivo. And finally, dual inhibition of MOZ/MORF and DOT1L exhibits better efficacy. Overall, this is a well-executed study elucidating the detailed epigenetic mechanism of AF-10 fusion proteins in target gene expression and leukemia cell survival and the results provide new therapeutic options for treating leukemias with AF10-fusions. Methodology is sound, and the conclusions are largely supported by the results. The paper is suitable for publishing with the questions below addressed.

(1) The authors claim that MOZ and ENL form a protein complex, but evidence supporting this claim seems insufficient. More evidence is needed to show they do form a stable protein complex or they may be simply proteins interacting with each other, or the authors need to tune down their claims.

(2) The model (Fig. 8) suggests that ENL and MOZ recruits AF10 fusions. The evidence used to support this claim is from an irrelevant 293T cells (Fig. 3). To support their model, these ChIP experiments need to be done in the myeloid progenitors as in Fig. 1 +/- AF10-fusions. In addition, it is better to assess genome-wide distributions of ENL, MOZ and AF-10 fusions in these cells to see whether the model proposed in Fig. 8 is a general model for most AF10 target genes or only for HOXA genes.

(3) The order of recruitment of MOZ, ENL and AF-10 fusions to target genes seems unclear, as depletion of each factor seems to affect the occupancy of others (Figs. 2, 3,5,7). This need to be reflected by the model in Fig. 8.

(4) In the model, other subunits of the MOZ complex (such as BRPF1) are included. Is it known that MOZ functions independent of its complex in AF-fusion mediated gene expression?

(5) Fig. 4, the MOZ/MORF dependency seems not specific to CALM-AF10 fusion cells, the MLL-ENL fusion cells are similarly sensitive to MOZ or MORF depletion. Also, since MOZ and MORF are somewhat complementary, double KO of both enzymes may give better results. Is there a possibility that like HBO1, MOZ/MORF are generally required for the survival of leukemia stem cells?

Minor:

(6) Fig. 2A, Venn diagrams or heatmaps would be better to assess the overlaps between the chromatin occupancies of ENL with other factors.

(7) Fig. 5, 10 uM of compounds are a bit too high that the cell toxicity may be due to off-target effect. A lower dose (such as 1-2 uM) should be included.

Reviewer #2 (Remarks to the Author):

In this manuscript, Komata et al. demonstrate that leukemia-associated fusion proteins involving the AF10 C-terminal fusion partner gene (such as MLL-AF10 and CALM-AF10) require the ENL-interacting OM-LZ domain in AF10 for their leukemic properties. Similar to MLL-ENL, direct linkage of ENL to the N-termini of CALM (or only the NES within the CALM N-terminus), NUP98 or DDX3X was sufficient to generate fusion proteins with leukemogenic properties. In turn, ENL was required for the interaction of fusion proteins with MOZ/MORF acetyltransferases. ChIP-seq studies showed that ENL was co-localized with MOZ and DOT1L on chromatin. Further structure-function studies showed that both YEATS and AHD domains of ENL were essential in the context of fusion-protein driven transformation of HSPC, chromatin binding and HOX gene expression. CRISPR/Cas9-

induced gene disruption showed that MOZ was required for the proliferation of CALM-AF10 cells while MORF appeared dispensable. A small molecule inhibitor of MOZ/MORF induced growth arrest in CALM-AF10 cells and reduced chromatin association of the fusion protein, ENL and MOZ and caused reduced HOX gene expression. MOZ/MORF inhibition showed robust antileukemic properties in vivo when CALM-AF10 murine and human leukemia cells were transplanted. Finally, dual inhibition of MOZ/MORF and the H3K79 methyltransferase DOT1L had a stronger antiproliferative effect.

This manuscript features a thorough and well-controlled structure-function analysis of cofactors that are required for leukemic fusion proteins that involve AF10. The finding that AF10-fusion containing leukemias are sensitive to MOZ/MORF inhibition could have important clinical implications. There are several open points that should be addressed in a revision:

- What could explain the differences in leukemia latency shown in Figure 1C?
- The authors should show more data to illustrate similarities/dissimilarities in RNA-seq profiles of the models shown in Figure 1D. What are conserved target genes / pathways that are induced by AF10-containing fusions? Supplementary Figure 2 does not contain the data as referred to in lines 125 and 127.
- The finding that fusion of ENL to minimal modules that greatly differ in their structural architecture cause HSPC transformation is very interesting. What could the structural/mechanistic commonalities between the MLL MTMT/CALM NES/NUP98/DDX3X be that mediate this effect?
- Why did the authors perform most ChIP-seq experiments in HEK293T cells? It would be much more relevant to show more data from hematopoietic cells. The analyses of the ENL ChIP-seq results shown in Figure 7A should be expanded to provide a global overview of ENL chromatin binding in leukemia cells.
- The authors claim that MOZ-deficiency causes reduced DOT1L/ENL chromatin recruitment based on the analysis of three target genes. This statement should be substantiated by testing more target genes or by performing ChIP-seq.
- What could be the reason for different MOZ/ENL/DOT1L binding to promoters vs. gene bodies? There appear to be some inconsistencies in the binding patterns of these factors between data shown in Figures 2D, 3E and 5E/G. While levels of MOZ/ENL and DOT1L are higher in the gene bodies than in the respective promoters in Figure 2D, the opposite is observed in Figure 5.
- Can the authors rule out that AF10 fusions form different ENL-containing subcomplexes that contain either DOT1L or MOZ?
- Figure 4: The data showing the differential sensitivity of CAML-AF10 and MLL-ENL-driven cells to MOZ knockout is weak. The authors should use more gRNAs to substantiate these findings. Assessment of indel formation by different gRNAs would also contribute to the robustness of this dataset.
- The interaction of MOZ/MORF and DOT1L inhibitors is interesting. Can the authors show that the drugs work in a synergistic fashion through isobologram analysis or other tests of drug synergy?
- Do MOZ/MORF and DOT1L inhibitors cooperatively reduce chromatin binding of CALM-AF10? What are the effects of double drug treatment on gene expression?
- To show that MOZ/MORF inhibition (potentially in combination with DOT1L inhibition) has clinical relevance, the authors should test effects of this treatment on primary human AF10-fusion-expressing leukemia cells vs. normal human bone marrow cells. IN addition, more leukemia cell lines with other cancer drivers should be tested.
- More data should be provided to substantiate the efficacy of the dual WM1119/EPZ-5676 treatment in vivo. What is the effect of this treatment on disease progression and overall survival?

- How do MLL-AF10-driven cells respond to MOZ/MORF inhibition?
- The amount of charts together with the color coding make it almost impossible to spot differences in proliferation in Figure 6D. This representation should be improved.
- Line 262: Cellular differentiation is not shown in Figure 6D.
- Given the interaction of the CALM NES with XPO1 and the importance of XPO1 for other fusion proteins (as mentioned in the discussion): Would an XPO1-ENL fusion also transform HSPCs?

RESPONSE TO REVIEWERS' COMMENTS

Reviewer #1 (Remarks to the Author):

The manuscript by Komata et al describes a role of MOZ and ENL in AF10 fusion proteins-mediated gene expression in leukemia. Using an ex vivo myeloid progenitor transformation assay and artificial gene constructs encoding various combined essential protein domains, the authors show that ENL is essential in AF10-fusions-mediated leukemic transformation. ENL occupies promoters of target genes that were also occupied by the MOZ, DOT1L, AEP, and HBO1 complexes, likely forming a positive feedback regulatory loop. Genetic ablation or pharmacological inhibition of the MOZ/MORF KATs suppresses CALM-AF10-mediated leukemia growth both in vitro and in vivo. And finally, dual inhibition of MOZ/MORF and DOT1L exhibits better efficacy. Overall, this is a well-executed study elucidating the detailed epigenetic mechanism of AF-10 fusion proteins in target gene expression and leukemia cell survival and the results provide new therapeutic options for treating leukemias with AF10-fusions.

Methodology is sound, and the conclusions are largely supported by the results. The paper is suitable for publishing with the questions below addressed.

(1) The authors claim that MOZ and ENL form a protein complex, but evidence supporting this claim seems insufficient. More evidence is needed to show they do form a stable protein complex or they may be simply proteins interacting with each other, or the authors need to tune down their claims.

Response: We reported that MOZ and ENL interact with each other as evidenced by co-purification analysis and reciprocal co-IP experiments in our previous study¹. However, previous studies done by others using conventional biochemical purification procedures² did not find ENL in MOZ/MORF complexes. We suspect that ENL is an interacting partner for MOZ that binds in a context-dependent manner at sub-stoichiometric levels. ENL is a component for many other transcriptional/epigenetic factors (e.g., DOT1L, AEP, PRC1)³. Thus, the exchange of ENL among different protein complexes likely occurs dynamically. In case of MOZ-ENL interaction, we think they do form a transitory complex, and contribute the ENL subunit to other complexes such as the DOT1L complex and AEP. Thus, ENL is likely a context-dependent interactor for MOZ/MORF, but not a component of a stable complex. We mentioned this perspective in the revised manuscript as follows in page 3.

“ENL was not found as a complex component of MOZ/MORF in the early biochemical purifications of stably assembled complexes², suggesting that ENL is a transient interactor that binds to MOZ/MORF in a context-dependent manner¹”

(2) The model (Fig. 8) suggests that ENL and MOZ recruits AF10 fusions. The evidence used to support

this claim is from an irrelevant 293T cells (Fig. 3). To support their model, these ChIP experiments need to be done in the myeloid progenitors as in Fig. 1 +/- AF10-fusions. In addition, it is better to access genome-wide distributions of ENL, MOZ and AF-10 fusions in these cells to see whether the model proposed in Fig. 8 is a general model for most AF10 target genes or only for HOXA genes.

Response: Successful ChIP experiments demand good antibodies and sufficient protein expression. We have tested all available antibodies and other techniques including CUT & TAG or CUT & RUN to detect ENL, MOZ, and AF10-fusions in hematopoietic cell lines, but did not find any experimental conditions/materials that successfully detected them, except for ENL in P31/FUJ cells, which is a myeloid leukemia cell line carrying the CALM-AF10 fusion. Thus, we performed ChIP-seq on ENL in Fig. 7a as a representative molecule. In P31/FUJ cells, ChIP-signals of endogenous MOZ and CALM-AF10 proteins could not be detected by the available antibodies, likely due to their relatively low protein expression (Supplementary Fig.8c). We agree with Reviewer#1 that it would be ideal to detect ENL, MOZ, and CALM-AF10 in the same hematopoietic cells. However, due to the technical constraints, the ENL ChIP in P31/FUJ leukemia cells was the only option. Hence, we monitored the ENL ChIP signals with or without MOZ/MORF inhibitors (Fig.7c). As a cofactor of ENL, we were able to detect ING4 (a MOZ/MORF complex component) and Cyclin T1 (an AEP component) by ChIP-qPCR, and therefore added the data in Fig. 7c. ENL ChIP-qPCR data in Fig.7c indicated that the presence of ENL at *HOXA5* and *HOXA9* is highly dependent on MOZ/MORF-HAT activity. Many leukemic oncogenes including *MYC*, *MYB*, *MEIS* were also downregulated by the MOZ/MORF HAT inhibitor (Fig.7b), and the presence of ENL was reduced as well (Fig.7c), indicating that at least, these oncogenes are likely regulated in the proposed model in leukemia cells.

HEK293T cells are a kidney-derived cells in which ENL-MOZ-DOT1L-mediated gene regulation is active, which makes them an ideal tool to study the physiological function of the ENL-MOZ-DOT1L axis. We showed here that NES-ENL activated *HOXA5* and *HOXA9* in HEK293T cells, indicating that a common transcriptional activation mechanism is used for *HOXA* genes in both P31/FUJ and HEK293T cells. The protein expression levels of MOZ and ENL in HEK293T cells were much higher than those in P31/FUJ cells (Supplementary Fig.8c), which makes it possible to detect the ChIP signals of those proteins in HEK293T cells. Moreover, HEK293T cells express exogenously transduced cells very efficiently, which makes it possible to detect FLAG-tagged CALM-AF10 in ChIP-qPCR analysis (Figs.3e and 5g). Thus, HEK293T cells were the only option with which we could monitor the genomic distribution of ENL, MOZ and CALM-AF10 in the same cell. The genome-wide ChIP-seq analysis of ENL in wildtype HEK293T and its MOZ-KO cells indicated that not all the ENL-occupied loci are MOZ-dependent, while *HOXA* loci are uniquely dependent on MOZ (Fig.2d-f). Thus, we chose to examine the localizations of these proteins at the *HOXA5* and *HOXA9* loci in HEK293T cells. We think the proposed model can be applied to *HOXA* and

other leukemic genes tested, but not all of the ENL-target genes. We added this notion in the revised text as follows in page 10.

“Reduced ENL ChIP signal was seen at a subset of ENL target genes including *HOXA* genes (Fig.2d, e, and Supplementary Fig. 3d). Furthermore, ChIP signals of DOT1L were severely reduced at the *HOXA* loci but its overall genome-wide localization was less drastically affected by MOZ knockout compared to ENL (Fig.2f). These results suggest that *HOXA* genes, but not all ENL-target genes, are regulated dependently on MOZ. “.

(3) The order of recruitment of MOZ, ENL and AF-10 fusions to target genes seems unclear, as depletion of each factor seems to affect the occupancy of others (Figs. 2, 3,5,7). This need to be reflected by the model in Fig. 8.

Response: In Fig. 2g, the presence of DOT1L and ENL was severely reduced by the depletion of MOZ. This indicates that MOZ reaches the target chromatin prior to DOT1L and ENL. Only ENL, not DOT1L, interacts with MOZ¹. Therefore, ENL binds to chromatin as a MOZ/ENL complex, which recruits DOT1L (Fig. 9). The presence of ENL was reduced by DOT1L depletion probably because ENL is in a stable complex with DOT1L, which is substantially responsible for the retention of ENL in chromatin (Fig.2g). Thus, we think that the primary order of recruitment is MOZ/ENL complex followed by DOT1L/CALM-AF10 complex. However, the presence of MOZ was reduced to some extent by DOT1L depletion (Fig.2g) or by DOT1L inhibition (Fig.5e). Thus, as Reviewer#1 pointed out, it is likely that each factor affects the occupancy of the others to some extent. The most plausible scenario is a transcriptional cycle model. Activating the transcription propels further recruitment of these factors by expediting this transcriptional activation cycle. We reflected that by inserting an arrow to indicate the feedback loop in Fig. 9 of the revised manuscript.

(4) In the model, other subunits of the MOZ complex (such as BRPF1) are included. Is it know that MOZ functions dependent of independent of its complex in AF-fusion mediated gene expression?

Response: To answer this question, we tested the dependency on BRPF1 in CRIPSR/Cas9-mediated dropout assay in Supplementary Fig.5. Because CALM-AF10-immortalized cells were dependent on BRPF1, it is likely that they are dependent on the MOZ complex including BRPF1.

(5) Fig. 4, the MOZ/MORF dependency seems not specific to CALM-AF10 fusion cells, the MLL-ENL fusion cells are similarly sensitive to MOZ or MORF depletion. Also, since MOZ and MORF are somewhat complimentary, double KO of both enzymes may give better results. Is there a possibility that like HBO1,

MOZ/MORF are generally required for the survival of leukemia stem cells?

Response: We have tested more sgRNAs for *Moz* and *Morf* in CRISPR/Cas9-mediated dropout assay in Supplementary Fig.5, and both of the cell lines were dependent on *Moz*, but not on *Morf*. Therefore, we changed the description accordingly. However, the tendency of CALM-AF10-immortalized cells (ICs) to stop proliferation more quickly than MLL-ENL-ICs was consistent. Despite the fact that both CALM-AF10- and MLL-ENL-ICs are dependent on MOZ, the sensitivity to WM1119 was strikingly different between CALM-AF10- and MLL-ENL-ICs (Fig.5b, Supplementary Fig.7b,c). These results support that CALM-AF10 leukemia cells are more dependent on the MOZ/MORF HAT activity than MLL-ENL leukemia cells are. *Hbo1* was shown to be more generally required for the survival of leukemia stem cells⁴. It should be noted that many leukemia cells lines such as MV4-11 cells and HB1119 cells, showed some dependency on MOZ/MORF HAT activity for their proliferation, but were not induced complete differentiation by its inhibition (Fig.5b, and Supplementary Fig. 6a, b). It is possible that the HBO1 protein may be generally required for the survival of leukemia stem cells, but its HAT activity may not. We mentioned this in the revised manuscript as follows in page 15.

“These results indicate that MLL leukemia cells are dependent on the presence of MOZ/MORF/HBO1 proteins but not their KAT activity.”

Minor:

(6) *Fig. 2A, Venn diagrams or heatmaps would be better to assess the overlaps between the chromatin occupancies of ENL with other factors.*

Response: According to the Reviewer#1's suggestion, we provided Venn diagrams in Supplementary Fig. 3c.

(7) *Fig. 5, 10 uM of compounds are a bit too high that the cell toxicity may be due to off-target effect. A lower dose (such as 1-2 uM) should be included.*

Response: In Fig. 8a, lower doses including 0.01, 0.1, and 1µM were tested and 1µM has shown growth retardation effects.

Reviewer #2 (Remarks to the Author):

In this manuscript, Komata et al. demonstrate that leukemia-associated fusion proteins involving the AF10 C-terminal fusion partner gene (such as MLL-AF10 and CALM-AF10) require the ENL-interacting OM-LZ domain in AF10 for their leukemic properties. Similar to MLL-ENL, direct linkage of ENL to the N-termini of

CALM (or only the NES within the CALM N-terminus), NUP98 or DDX3X was sufficient to generate fusion proteins with leukemogenic properties. In turn, ENL was required for the interaction of fusion proteins with MOZ/MORF acetyltransferases. ChIP-seq studies showed that ENL was co-localized with MOZ and DOT1L on chromatin. Further structure-function studies showed that both YEATS and AHD domains of ENL were essential in the context of fusion-protein driven transformation of HSPC, chromatin binding and HOX gene expression. CRISPR/Cas9-induced gene disruption showed that MOZ was required for the proliferation of CALM-AF10 cells while MORF appeared dispensable. A small molecule inhibitor of MOZ/MORF induced growth arrest in CALM-AF10 cells and reduced chromatin association of the fusion protein, ENL and MOZ and caused reduced HOX gene expression. MOZ/MORF inhibition showed robust antileukemic properties in vivo when CALM-AF10 murine and human leukemia cells were transplanted. Finally, dual inhibition of MOZ/MORF and the H3K79 methyltransferase DOT1L had a stronger antiproliferative effect.

This manuscript features a thorough and well-controlled structure-function analysis of cofactors that are required for leukemic fusion proteins that involve AF10. The finding that AF10-fusion containing leukemias are sensitive to MOZ/MORF inhibition could have important clinical implications. There are several open points that should be addressed in a revision:

- *What could explain the differences in leukemia latency shown in Figure 1C?*

Response: MLL-AF10 is known as a very strong oncogenic driver in murine leukemia models. Compared to MLL-AF10, NES-AF10 or NES-ENL were slow or inefficient in developing leukemia, suggesting that these NES fusions are relatively weaker oncogenic drivers compared to MLL-AF10. It is unclear what makes this difference at this point. Requirements for specific cell-of-origin, additional mutations, or adaptation to environment might be the factors that potentially explain the difference of leukemia latency. One potential explanation could be that MLL-AF10 more efficiently binds to its target chromatin compared to NES-AF10 and NES-ENL, making MLL-AF10 a more stronger oncogenic driver. MLL-AF10 can bind to CpG-rich promoters via the CXXC domain of MLL, whereas NES-AF10/NES-ENL rely on YEATS-mediated interaction to acetylated nucleosomes to localize on the target genes.

- *The authors should show more data to illustrate similarities/dissimilarities in RNA-seq profiles of the models shown in Figure 1D. What are conserved target genes / pathways that are induced by AF10-containing fusions? Supplementary Figure 2 does not contain the data as referred to in lines 125 and 127.*

Response:

We examined the differentially expressed genes (DEGs) in RNA-seq profiles between the c-Kit-positive BM and AF10 fusion ICs. As shown Supplementary Fig.2d, e, pathways involved in lysosome, hemopoiesis, and lipid biosynthesis are commonly affected by AF10 fusions. We also examined the DEGs between DDX3X- and NUP98-AF10 fusions-ICs VS CALM- and MLL-AF10 fusions-ICs. Supplementary Fig. 2f,g, shows pathways involved in the MAPK cascade, protein kinase C signaling, and regulation of kinase activity are differentially affected. We added the text as follows in page 8.

“Other differentially expressed genes in c-Kit⁺-BM and AF10 fusion-ICs are the genes implicated in lysosome, hemopoiesis, and lipid biosynthesis (Supplementary Fig. 2d, e). CALM-AF10- and NES-ENL-ICs tended to have near-identical expression profiles, which is fairly similar to those of MLL-AF10-ICs, while NUP98-AF10⁻ and DDX3X-AF10⁻ ICs expressed slightly different gene sets, which included the genes implicated in MAPK cascades, suggesting that the activation status of proliferative signaling pathways may be different among the AF10 fusion subtypes (Supplementary Fig. 2c, f, g). “

It was our error to type Supplementary Fig. 2 in the previous version of this manuscript, which is corrected in the revised manuscript

• *The finding that fusion of ENL to minimal modules that greatly differ in their structural architecture cause HSPC transformation is very interesting. What could the structural/mechanistic commonalities between the MLL MTMT/CALM NES/NUP98/DDX3X be that mediate this effect?*

Response: As for MLL-AF10, the minimum structural requirements are the minimum targeting module (MTM) and the THD2 domain derived from the MLL portion. These structures confer binding to CpG-rich promoters as well as the ability to recruit HBO1^{3,5}. THD2-mediated HBO1 recruitment should promote subsequent AEP loading to the chromatin⁵. Because MLL-AF10 binds to its target chromatin without YEATS domain, the YEATS domain-mediated chromatin targeting is not required for MLL-AF10³. On the other hand, CALM-AF10 requires the YEATS domain-mediated chromatin binding in order to be tethered to chromatin (Fig.3e). NUP98 and DDX3X also required YEATS-domain mediated chromatin binding for oncogenic transformation (unpublished data). NES likely promotes localization of CALM-AF10 complex to the transcriptional factory via XPO1 (Fig.9). XPO1 may be commonly used to activate transcription by non-MLL-rearranged AF10 fusions. Thus, MLL-AF10 fusion and the other non-MLL-rearranged AF10 fusions differ in their mechanisms of chromatin targeting and transcriptional activation. The latter requires YEATS-domain-mediated chromatin association as a primary mechanism to bind chromatin. However, AEP-mediated gene activation is likely the mechanism commonly employed by AF10 fusions.

• *Why did the authors perform most ChIP-seq experiments in HEK293T cells? It would be much more relevant to show more data from hematopoietic cells. The analyses of the ENL ChIP-seq results shown in*

Figure 7A should be expanded to provide a global overview of ENL chromatin binding in leukemia cells.

Response: As the Reviewer #2 suggested, It is ideal to perform these ChIP-seq analysis on leukemia cells. However, as we mentioned in the response to the Reviewer#1, we can only detect the genomic localization of ENL, but not MOZ, and CALM-AF10 in P31/FUJ leukemia cells likely due to their relatively low protein expression levels (Supplementary Fig.8c). To detect ENL, MOZ, DOT1L and CALM-AF10 in the same cell, HEK293T cells are the only available option at this moment, because HEK293T cells express those proteins at much higher levels. We showed here that NES-ENL activated *HOXA5* and *HOXA9* in HEK293T cells, which indicated that NES-ENL-mediated gene activation can be monitored in HEK293T cells as well as in hematopoietic cells. Thus, we analyzed the mechanism of *HOXA* gene activation in HEK293T cells using ChIP analysis to study the molecular mechanism of ENL-mediated gene expression.

ENL is localized around the TSS of genes both in P31/FUJ and HEK293T cells in the genome-wide manner (Supplementary Fig.8a,b). P31/FUJ cells expressed ENL at lower levels compared to HEK293T cells (Supplementary Fig.8c). As a result, the localization of ENL at promoter-proximal cording regions was relatively low in P31/FUJ cells compared to HEK293T cells, accompanied with lower presence of RNAP2 Ser5-P in the cording regions in P31/FUJ cells. Those features of HEK293T cells, which express higher levels of MOZ, provided us an opportunity to analyze the relative localization of ENL, MOZ, and CALM-AF10 in the same cells (Fig. 5g).

• *The authors claim that MOZ-deficiency causes reduced DOT1L/ENL chromatin recruitment based on the analysis of three target genes. This statement should be substantiated by testing more target genes or by performing ChIP-seq.*

Response: According to Reviewer #2's suggestion, we have performed ChIP-seq analysis on ENL and DOT1L in MOZ-deficient HEK293T cells. It turned out that the presence of ENL was not uniformly dependent on MOZ in all ENL-occupied genes (Fig.2e). *HOXA* genes were uniquely dependent on the presence of MOZ. Thus, we used HEK293T cells to explore the molecular mechanisms of *HOXA*-gene regulation by the MOZ-ENL-DOT1L axis as a model case in this study and changed the description accordingly.

• *What could be the reason for different MOZ/ENL/DOT1L binding to promoters vs. gene bodies? There appear to be some inconsistencies in the binding patterns of these factors between data shown in Figures 2D, 3E and 5E/G. While levels of MOZ/ENL and DOT1L are higher in the gene bodies than in the respective promoters in Figure 2D, the opposite is observed in Figure 5.*

Response: As the Reviewer #2 mentioned, there is an instability in the binding patterns to promoters compared to gene bodies. At this point, we don't know the reason. The binding patterns may be somewhat dynamic depending on the culture conditions. For fair comparison, we have performed experiments by culturing the cells side-by-side and prepared the chromatin samples in the same condition.

• *Can the authors rule out that AF10 fusions form different ENL-containing subcomplexes that contain either DOT1L or MOZ?*

Response: ENL binds MOZ independently on DOT1L/AF10 complexes, as evidenced by ENL-MOZ interaction in the absence of DOT1L (Supplementary Fig.4b). Thus, we think that MOZ/ENL complex and DOT1L/AF10/ENL complex are different entities. We speculate that AF10 fusions mainly form a relatively stable AF10 fusion/DOT1L/ENL complex, but not with MOZ. However, we think it's possible that AF10 fusions form a AF10 fusion/ENL/MOZ complex at least in a transient manner as depicted in Fig. 9. It is unlikely that AF10 fusion forms a complex with MOZ in the absence of DOT1L as AF10 fusions bind to ENL via DOT1L as shown in Fig.1b. Thus, we think that AF10 fusions can form an AF10 fusion/DOT1L/ENL complex or an AF10 fusion/DOT1L/ENL/MOZ complex (transiently) but they can't form an AF10 fusion/MOZ/ENL complex because they need DOT1L to form a complex with ENL.

• *Figure 4: The data showing the differential sensitivity of CAML-AF10 and MLL-ENL-driven cells to MOZ knockout is weak. The authors should use more gRNAs to substantiate these findings. Assessment of indel formation by different gRNAs would also contribute to the robustness of this dataset.*

Response: We generated more sgRNA for both *Moz* and *Morf* to substantiate our findings as suggested by the Reviewer#2. We have tested more sgRNA for *Moz* and *Morf* in CRIPSR/Cas9-mediated dropout assay in Supplementary Fig.4, and both of the cell lines are dependent on *Moz* but not *Morf*. Thus, MLL-ENL-ICs are also dependent on *Moz* in contrast to our initial interpretation. However, the tendency of CALM-AF10-ICs to stop proliferation more quickly than MLL-ENL-ICs was consistent. We changed the description accordingly as follows in page 13.

“*Moz* knockout quickly eliminated CALM-AF10-ICs, while it also attenuated the proliferation of MLL-ENL-ICs. *Morf* knockout did not dramatically affect the proliferation of CALM-AF10-ICs or MLL-ENL-ICs, suggesting that MOZ is the predominant ortholog in these cell types. These tendencies were consistently observed using multiple different sgRNAs (Supplementary Fig.5). Moreover, *Brpf1*, a common component of MOZ/MORF complexes, was also required for both cell types.”

• *The interaction of MOZ/MORF and DOT1L inhibitors is interesting. Can the authors show that the drugs*

work in a synergistic fashion through isobologram analysis or other tests of drug synergy?

Response: The inhibitors for MOZ/MORF HATs and DOT1L HMT induced differentiation at a slower kinetic rate compared to other agents that rapidly induce apoptosis. Thus, we relied on growth curve analysis to detect the difference at day 12 and later. This makes it difficult to perform conventional isobologram analysis. We showed synergistic/additive effects of two inhibitors at growth curve analysis of two different cell lines (Fig.8a, Supplementary Fig. 9a), FACS analysis (Fig.8b), gene expression analysis (Fig.8c), and colony formation analysis in semi-solid medium (Fig.8d,e).

• Do MOZ/MORF and DOT1L inhibitors cooperatively reduce chromatin binding of CALM-AF10? What are the effects of double drug treatment on gene expression?

Response: Although we could not monitor the chromatin binding of CALM-AF10 in leukemia cells for the aforementioned reason, the effects on gene expression by two inhibitors were examined in Fig.8c. Leukemic oncogenes such as *MYC*, *MYB*, *MEIS1*, *HOXA5* and *HOXA9* were cooperatively downregulated by WM1119 and EPZ5676.

• To show that MOZ/MORF inhibition (potentially in combination with DOT1L inhibition) has clinical relevance, the authors should test effects of this treatment on primary human AF10-fusion-expressing leukemia cells vs. normal human bone marrow cells. IN addition, more leukemia cell lines with other cancer drivers should be tested.

Response: According to the Reviewer #2's suggestion, we tested the effects of single or double inhibitors on normal CD34+ human hematopoietic cells in colony formation assay. As shown in Fig.8d,e, the cell numbers and colony numbers of normal CD34+ cells were not drastically altered by the drug treatment, where those of P31/FUJ cells were greatly affected. We also tested whether CALM-AF10 transformed human primary hematopoietic cells, but the introduction of CALM-AF10 gene in human CD34+ cells did not immortalize in our culture conditions. Therefore we tested additional leukemia cell line called KP-Mo-TS for drug sensitivity (Supplementary Fig.9a), which exhibited a similar tendency to P31/FUJ cells.

• More data should be provided to substantiate the efficacy of the dual WM1119/EPZ-5676 treatment in vivo. What is the effect of this treatment on disease progression and overall survival?

Response: The disease progression and overall survival is shown in the Fig.8f.

- *How do MLL-AF10-driven cells respond to MOZ/MORF inhibition?*

Response: The response of MLL-AF10-ICs to MOZ/MORF inhibition is shown in Supplementary Fig7a. Overall, MLL-AF10-ICs are less sensitive to MOZ/MORF inhibition compared to other AF10 fusions-ICs

- *The amount of charts together with the color coding make it almost impossible to spot differences in proliferation in Figure 6D. This representation should be improved.*

Response: We reorganized the figures and show the growth curve of each cell line (biological replicates) in the revised manuscript (Fig.6b, Supplementary Fig.7a)

.

- *Line 262: Cellular differentiation is not shown in Figure 6D.*

Response: Cellular differentiation was evaluated by the expression levels of CD11b in the revised Fig.6c.

- *Given the interaction of the CALM NES with XPO1 and the importance of XPO1 for other fusion proteins (as mentioned in the discussion): Would an XPO1-ENL fusion also transform HSPCs?*

Response: As suggested by the Reviewer#2, we generated XPO1-ENL fusion and tested its transforming ability. As predicted, XPO1-ENL fusion activated *Hoxa9* expression and immortalized hematopoietic progenitors (Fig.3a), which support our model (Fig.9)

References

- 1 Miyamoto, R. *et al.* Activation of CpG-Rich Promoters Mediated by MLL Drives MOZ-Rearranged Leukemia. *Cell Rep* **32**, 108200, doi:10.1016/j.celrep.2020.108200 (2020).
- 2 Doyon, Y. *et al.* ING tumor suppressor proteins are critical regulators of chromatin acetylation required for genome expression and perpetuation. *Mol Cell* **21**, 51-64, doi:10.1016/j.molcel.2005.12.007 (2006).
- 3 Okuda, H. *et al.* Cooperative gene activation by AF4 and DOT1L drives MLL-rearranged leukemia. *J Clin Invest* **127**, 1918-1931, doi:10.1172/JCI91406 (2017).
- 4 MacPherson, L. *et al.* HBO1 is required for the maintenance of leukaemia stem cells. *Nature* **577**, 266-270, doi:10.1038/s41586-019-1835-6 (2020).

- 5 Takahashi, S. *et al.* HBO1-MLL interaction promotes AF4/ENL/P-TEFb-mediated leukemogenesis. *Elife* **10**, doi:10.7554/eLife.65872 (2021).

REVIEWERS' COMMENTS

Reviewer #1 (Remarks to the Author):

The authors have adequately addressed my questions with experimental data and textural changes. Therefore I recommend the paper for publication.

Reviewer #2 (Remarks to the Author):

In the revised version of the manuscript, the authors have sufficiently responded to the points I raised before, and the newly included results are of very good quality.

I have only one small remark to Figure 2E/F:

The graphical representation that was chosen to show differential chromatin association of ENL and DOT1L in ctrl vs. MOZ-deficient cells does not fully support the statement in lines 155-160, as the plots in Figure 2E/F do not allow to make a quantitative assessment of this effect. Therefore, I suggest that the authors choose a different way to represent these data.

If this point is addressed the manuscript can be accepted for publication.

RESPONSE TO REVIEWERS' COMMENTS

Reviewer #1 (Remarks to the Author):

The authors have adequately addressed my questions with experimental data and textural changes. Therefore I recommend the paper for publication.

Response:

Dear Reviewer, thank you for your appreciative words. We are glad that our revisions in response to your comments were satisfactory. We thank you for your time and efforts in reviewing our manuscript and providing insightful suggestions.

Reviewer #2 (Remarks to the Author):

In the revised version of the manuscript, the authors have sufficiently responded to the points I raised before, and the newly included results are of very good quality.

I have only one small remark to Figure 2E/F:

The graphical representation that was chosen to show differential chromatin association of ENL and DOT1L in ctrl vs. MOZ-deficient cells does not fully support the statement in lines 155-160, as the plots in Figure 2E/F do not allow to make a quantitative assessment of this effect. Therefore, I suggest that the authors choose a different way to represent these data.

If this point is addressed the manuscript can be accepted for publication.

Response:

Dear Reviewer, thank you for your kind words. We are glad that our revisions in the previous round in response to your comments were satisfactory. We appreciate your time and efforts in reviewing our revised manuscript and highlighting this important point.

Following your suggestion, we changed Figure 2e and f in the revised manuscript to ensure a better presentation. The point we want to present here is that the presence of ENL and DOT1L is uniquely down-regulated at the HOXA locus in MOZ-deficient cells. We extracted the top 100 genes that were bound by ENL (e) or DOT1L (f) and plotted their relative ChIP signals of ENL (e) or DOT1L (f). In both cases, HOXA6 ranked as the most severely reduced gene. We also changed the text accordingly as follows:

“Reduced ENL ChIP signal was uniquely seen at a subset of ENL target genes including *HOXA* genes (Fig.2e, and Supplementary Fig. 3d). Furthermore, ChIP signals of DOT1L were severely reduced at the *HOXA* loci as well by MOZ knockout (Fig.2f, and Supplementary Fig. 3d). These results suggest that *HOXA* genes, but not all ENL-target genes, are regulated dependently on MOZ.”